# MEMOIR: Lifelong Model Editing with Minimal Overwrite and Informed Retention for LLMs

**Ke Wang**[*]
EPFL, Lausanne, Switzerland
k.wang@epfl.ch

**Yiming Qin**[*]
EPFL, Lausanne, Switzerland
yiming.qin@epfl.ch

**Nikolaos Dimitriadis**
EPFL, Lausanne, Switzerland
nikolaos.dimitriadis@epfl.ch

**Alessandro Favero**
EPFL, Lausanne, Switzerland
alessandro.favero@epfl.ch

**Pascal Frossard**
EPFL, Lausanne, Switzerland
pascal.frossard@epfl.ch

## Abstract

Language models deployed in real-world systems often require post-hoc updates to incorporate new or corrected knowledge. However, editing such models efficiently and reliably—without retraining or forgetting previous information—remains a major challenge. Existing methods for lifelong model editing either compromise generalization, interfere with past edits, or fail to scale to long editing sequences. We propose MEMOIR, a novel scalable framework that injects knowledge through a residual memory, i.e., a dedicated parameter module, while preserving the core capabilities of the pre-trained model. By sparsifying input activations through sample-dependent masks, MEMOIR confines each edit to a distinct subset of the memory parameters, minimizing interference among edits. At inference, it identifies relevant edits by comparing the sparse activation patterns of new queries to those stored during editing. This enables generalization to rephrased queries by activating only the relevant knowledge while suppressing unnecessary memory activation for unrelated prompts. Experiments on question answering, hallucination correction, and out-of-distribution generalization benchmarks for LLaMA-3 and Mistral backbones demonstrate that MEMOIR achieves state-of-the-art performance across reliability, generalization, and locality metrics, scaling to thousands of sequential edits with minimal forgetting. [2]

## 1 Introduction

Large language models (LLMs) exhibit strong performance across a wide range of tasks, enabled by large-scale pre-training on diverse and extensive datasets [1, 2]. However, LLMs can generate outdated or inaccurate information and can perpetuate biases during deployment [3, 4, 5], which requires further continuous updates to their knowledge base after pre-training to correct these behaviors. Although fine-tuning on new corpora can revise model behavior, it is computationally expensive and prone to catastrophic forgetting [6, 7]. These limitations motivate an emerging line of work known as *lifelong model editing* [8, 9], which aims to continuously update the model's knowledge in an efficient and localized manner.

---

[*]Equal contribution.
[2]Code at https://github.com/qym7/MEMOIR

39th Conference on Neural Information Processing Systems (NeurIPS 2025).

In lifelong model editing, a model is updated sequentially with a stream of edits to continuously update its stored knowledge, each modifying the model's response to a *singular* input prompt such as "Where was the last Summer Olympics held?". To ensure that the model can faithfully produce corrected predictions on input queries relevant to all updated knowledge after editing, each edit has to be i) *reliable*, it should faithfully correct the model's response to the edited input; ii) *generalizable*, it should generalize to semantically similar prompts to the edited prompt such as "Which city held the last Summer Olympics?" iii) *localized*, it should minimally affect the model's responses to previously edited and irrelevant input prompts. Balancing these three criteria is the central challenge in designing an effective and efficient lifelong model editor.

Previous methods for knowledge editing can be broadly categorized into *non-parametric* and *parametric* approaches. Non-parametric methods [8, 10] store fixed input-output activation patterns to directly correct the model's predictions during inference. While this enables precise and localized knowledge edits, such rigid memorization typically generalizes poorly to semantically similar queries. In contrast, parametric approaches [11, 12, 13, 9, 14, 15] modify a subset of model parameters—either within the original network or in a dedicated memory module—to integrate new knowledge. Although these methods offer better generalization to related inputs, continuous editing can still cause newer updates to overwrite earlier ones, resulting in catastrophic forgetting of previous edited knowledge and limiting their reliability over long sequences of edits.

In this paper, we propose MEMOIR, (Model Editing with Minimal Overwrite and Informed Retention), which delivers a desirable balance between reliability, generalization, and locality for a large number of edits (Figure 1). MEMOIR introduces a *memory module*, i.e., a dedicated fully-connected layer in a single transformer block where all edits are performed. MEMOIR mitigates catastrophic forgetting by allocating distinct parameter subsets to each edit and retrieving them during inference to activate only the relevant knowledge for a given prompt. During editing, we propose a structured sparsification of the input with sample-dependent masks to this layer, dynamically activating only a sample-specific subset of parameters in the introduced memory module in the forward pass. MEMOIR

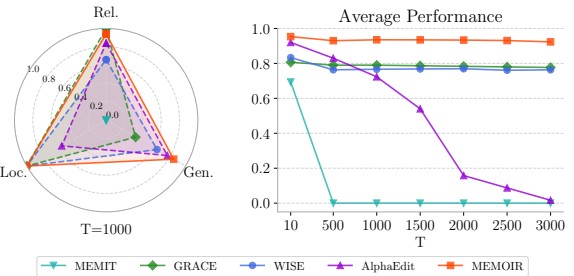

Figure 1: Left: trade-offs among reliability (Rel.), generalization (Gen.), and locality (Loc.) for 1,000 continual edits. Right: Average performance of Rel., Gen., and Loc. under varying numbers of edits. Both evaluated on LLaMA-3-8B-Instruct [16] with ZsRE dataset [17]. MEMOIR delivers the best balance among the three metrics and scales with a large number of edits.

therefore promotes the distribution of new knowledge across the parameter space, reducing the overwriting of previous updates and significantly mitigating catastrophic forgetting. During inference, we use the sparsification pattern of a prompt to infer if it semantically corresponds to an edited prompt, and route the activations accordingly. This targeted knowledge activation eliminates the need for large corpora of irrelevant samples to regularize training [14, 15, 9]. Specifically, if the input corresponds to a rephrased version of an edit, we activate only the relevant knowledge of that edit to align the representation of the prompt. Conversely, the introduced module is deactivated when detecting irrelevant prompts, effectively preserving the pre-trained knowledge of the LLM.

Our contributions are as follows:

- We propose MEMOIR, a novel lifelong model editing method to continuously edit a long sequence of samples with minimal forgetting via sparsely distributing knowledge in the memory module from each edit. During inference, MEMOIR identifies edited samples, their rephrased counterparts, and irrelevant samples to dynamically activate or deactivate relevant knowledge.

- We extensively evaluate MEMOIR on Q&A, hallucination correction, OOD generalization, and multi-hop reasoning tasks, demonstrating state-of-the-art results across LLaMA-3 [16], Mistral [18], LLaMA-2 [19] and GPT-J [20] architectures compared to all previous methods.

- We extend the previous edit horizon to 15,000 edits and show that `MEMOIR` consistently delivers superior performance in the challenging setting of sequential singular edits compared with previous editing methods.

## 2 Lifelong Model Editing

**Problem formulation** Lifelong model editing aims to continuously add new knowledge into a model without forgetting previously acquired knowledge [8, 9]. Let $\boldsymbol{\theta} \in \mathbb{R}^d$ be the parameters of an LLM denoted as a function $f_{\boldsymbol{\theta}} : \mathcal{X} \to \mathcal{Y}$. It maps an input prompt $\boldsymbol{x} \in \mathcal{X}$ to an output prediction $f_{\boldsymbol{\theta}}(\boldsymbol{x}) \in \mathcal{Y}$. The LLM has been trained with internet-scale corpora $\mathcal{D}_{\text{train}}$. During editing, the model receives a time-evolving stream of edit samples $\mathcal{D}_{\text{edit}} = \{(\boldsymbol{x}_e^t, \boldsymbol{y}_e^t)\}_t$, where each pair $(\boldsymbol{x}_e^t, \boldsymbol{y}_e^t) \in \mathcal{D}_{\text{edit}}$ represents the $t$-th edit sample in the edit sequence. In real-world settings, this sequence is expected to grow continually, potentially reaching thousands of edits.

When performing the $t$-th edit, the goal of lifelong model editing is to incorporate the new edit while preserving previously acquired knowledge. Formally, for the parameters of the $t$-th edit $\boldsymbol{\theta}^t$, our objective is for the updated model to predict $\boldsymbol{y} = f_{\boldsymbol{\theta}^t}(\boldsymbol{x})$, $\forall (\boldsymbol{x}, \boldsymbol{y}) \in \mathcal{D}_{\text{train}} \cup \mathcal{D}_{\text{edit}}^{\leq t}$ where $\mathcal{D}_{\text{edit}}^{\leq t} := \{(\boldsymbol{x}_e^s, \boldsymbol{y}_e^s)\}_{s \leq t}$. In practice, $\mathcal{D}_{\text{train}}$ is often unavailable and we approximate the retention of pre-trained knowledge by a predefined set of samples irrelevant to the edit dataset: $(\boldsymbol{x}, \boldsymbol{y}) \in \mathcal{D}_{\text{irr}}$.

**Knowledge editing for LLMs** LLMs are typically based on the transformer architecture [21] and consist of $L$ identical blocks containing multi-head attention modules followed by feed-forward network (FFN) modules. For each block $\ell \in [L]$, the FFN module consists of two fully connected layers, parameterized by $\boldsymbol{W}_{\text{fc}}^{\ell}$ and $\boldsymbol{W}_{\text{proj}}^{\ell}$, and an activation function. Previous work shows that knowledge is typically stored in the middle blocks of an LLM [11], and updating the parameters of a single middle block rather than fine-tuning the entire model can be effective to edit factual knowledge [9, 8]. In what follows, to lighten notation, we will omit the layer-indexing superscript. Let $\boldsymbol{h}$ represent the output from the attention module; then the output of the FFN module is formally defined as:

$$\text{FFN}(\boldsymbol{h}) = \boldsymbol{W}_{\text{proj}} \, \sigma(\boldsymbol{W}_{\text{fc}} \, \boldsymbol{h}), \tag{1}$$

for an element-wise activation function $\sigma(\cdot)$. The FFN module can be interpreted as a two-layer key-value memory system [22], where the second layer $\boldsymbol{W}_{\text{proj}}$ functions as a linear associative memory [11], mapping a sequence of input vectors $[\boldsymbol{a}_1, \boldsymbol{a}_2, \dots]$ to corresponding output vectors $[\boldsymbol{v}_1, \boldsymbol{v}_2, \dots]$. This memory-like behavior enables information retrieval, where $\boldsymbol{a} = \sigma(\boldsymbol{W}_{\text{fc}} \, \boldsymbol{h})$ and $\boldsymbol{v} = \text{FFN}(\boldsymbol{h})$ respectively represent the input and output vectors of the layer $\boldsymbol{W}_{\text{proj}}$. Leveraging this perspective, previous works have shown the effectiveness of modifying the output of $\boldsymbol{W}_{\text{proj}}$ to update the model's stored knowledge [11, 22, 9, 14, 15]. However, continuously updating $\boldsymbol{W}_{\text{proj}}$ in lifelong editing gradually overwrites previous edits, leading to significant forgetting and severe performance degradation with a large number of edits, as shown in Figure 1.

## 3 Memory Editing with Minimal Overwrite and Informed Retention

We propose `MEMOIR`: Model Editing with Minimal Overwrite and Informed Retention. During editing, `MEMOIR` distributes knowledge storage by allocating distinct parameter subsets for each edit to reduce overwriting of previous edits (Section 3.1). At inference stage, `MEMOIR` dynamically activates relevant parameters related to the knowledge of each input prompt (Section 3.2). Through this design, `MEMOIR` achieves a superior balance among reliability, generalization, and locality even with a large number of edits. We start by presenting the general framework of `MEMOIR` illustrated in Figure 2.

**Editing layer output with residual memory** Following prior works [11, 15], we solely modify the output of a projection layer of the FFN $\boldsymbol{W}_{\text{proj}}$ of a specific block $\ell$, which is denoted as $\boldsymbol{W}_0$ in the following. To preserve the knowledge stored in the pre-trained model, we do not directly modify the pre-trained weights of this layer but introduce a *residual memory layer* $\boldsymbol{W}_{\text{m}}$ to incorporate new edits. $\boldsymbol{W}_{\text{m}}$ is a zero-initialized copy of the original matrix $\boldsymbol{W}_0$ such that it contains no information before editing. The final output of the edited layer combines the output of the original layer and the residual memory layer.

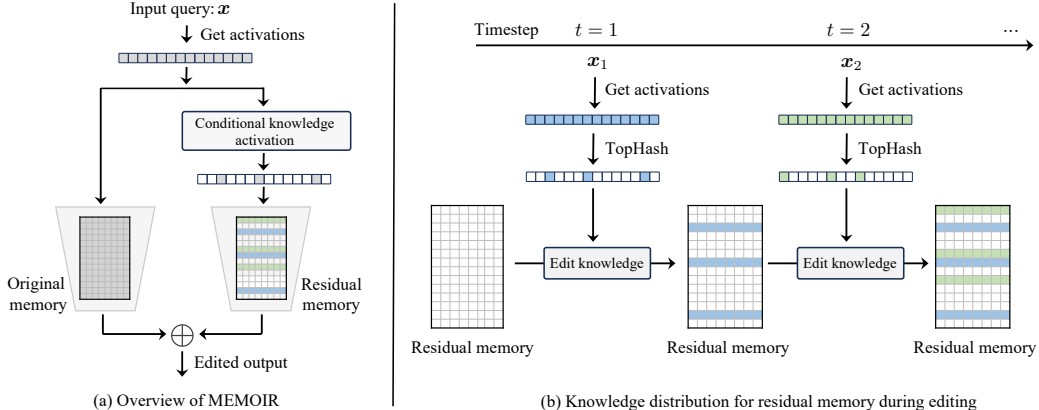

| (a) Overview of MEMOIR | (b) Knowledge distribution for residual memory during editing |

Figure 2: (a) Overall framework of MEMOIR during inference stage. The edited output combines the outputs of the original layer output and the residual memory. The input to the residual memory conditionally activates specific columns in the residual memory to retrieve relevant knowledge. (b) During editing, each edit modifies only a designated subset of columns in the residual memory, minimizing overwriting of previous edits in the memory. For visualization, we transpose the weight matrices.

## 3.1 Editing: distributing knowledge across memory

Prior parametric approaches on knowledge editing [14, 15] typically store the knowledge of each edit across the entire trainable parameter space. However, unconstrained updates on all trainable parameters lead to severe catastrophic forgetting, as new edits quickly overwrite knowledge from previous ones, as shown in Figure 1. In this work, we draw motivation from the continual learning literature leveraging sparsity [23, 24, 25, 26, 27, 28] to mitigate catastrophic forgetting, and propose a structured sparsification of input activations to the residual memory during the forward pass. This ensures that each edit modifies only a subset of the trainable parameters, promoting less overlap in representations and leading to more localized updates across different edits.

Concretely, for an input prompt $x$ let the input activations of the edited layer be $a(x) \in \mathbb{R}^D$, where we omit the token dimension for clarity. We apply a binary mask $\mathcal{M} : \mathbb{R}^D \mapsto \{0, 1\}^D$ to the input activations, and the final output of the edited layer, denoted as $\text{FFN}_{\text{edited}}(a(x))$, is calculated as:

$$\text{FFN}_{\text{edited}}(a(x)) = W_0 \, a(x) + W_{\text{m}} \, (\mathcal{M}(a(x)) \odot a(x)), \quad (2)$$

Applying the sparse mask $\mathcal{M}(a(x))$ to the input activations retains only $k \ll D$ *active indices*, setting the rest to zero. As a result, gradient updates are restricted to the $k$ corresponding columns in $W_{\text{m}}$, where the knowledge for input $x$ is explicitly stored. The other columns remain unchanged, effectively preserving previously stored edits and minimizing interference. In practice, we construct the mask based on the activation averaged across all tokens in the prompt $x$ and then apply it uniformly to the activation of each token. To suppress features with consistently high magnitude across all prompts, the average activations are centered by removing their mean computed with 100 irrelevant prompts. This enables a more diverse selection of masks across prompts. We provide more details and ablations in Appendix B.2. Next, we describe how the mask is derived using the TopHash mechanism.

**TopHash** A fundamental aspect of this sparse masking is the selection mechanism for the *active indices*. Intuitively, this mechanism should satisfy two different criteria: selection of feature diversity and semantic coherence. Diversity in feature selection encourages the model to consider a broader set of features, reducing the risk of overfitting on dominant features and mitigating catastrophic forgetting by distributing updates across the trainable parameters of the memory module. Semantic coherence ensures that semantically similar input prompts yield similar selected masks, facilitating the activation of relevant knowledge in the memory module for previously unseen yet semantically related queries. Therefore, we opt for *structured sparsity* [29].

The masking should be sample-dependent so that different feature subsets are chosen based on the input characteristics. A naive way to introduce diversity in feature selection is to generate a random seed for index selection directly from the input activations; however, this results in highly input-specific masks that fail to generalize to semantically similar prompts, such as rephrased edits,

defeating the purpose of structured and reusable knowledge storage. To address this limitation, we turn to a more robust fingerprinting mechanism based on the inherent structure of the activations themselves. For an input $\boldsymbol{a} = \boldsymbol{a}(\boldsymbol{x}) \in \mathbb{R}^D$, we compute a top-k selection based on feature magnitudes, mapping $\boldsymbol{a}$ to a binary mask $\mathcal{T}(\boldsymbol{a})$. Denoting as $a_{(k)}$ the $k$-th largest value using order statistics notation, the mask is formally defined as $\mathcal{T}(\boldsymbol{a})_j = \mathbb{1}[a_j \geq a_{(k)}],\ j \in [D]$. This strategy ensures that semantically similar input

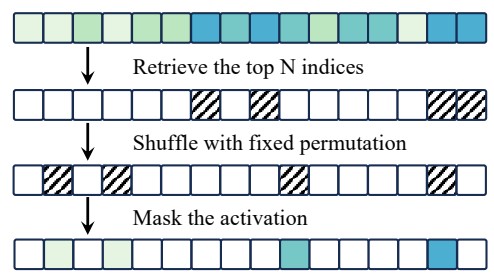

Figure 3: Illustration of TopHash.

prompts—naturally producing similar activation patterns—yield similar masks, directly addressing the generalization shortcoming of random seeds. Given the structured and highly correlated nature of activations in LLMs, this step effectively acts as a Locality-Sensitive Hashing (LSH) [30] mechanism, mapping semantically similar input prompts to identical or closely related binary codes [31].

Selecting the exact top-$k$ features concentrates updates on the most salient and frequently shared features across input prompts. Due to their large activation magnitudes, they are prone to being sensitive to weight changes and updates to the corresponding parameters increase the likelihood of interference with previously acquired knowledge and exacerbate catastrophic forgetting. To introduce diversity without sacrificing semantic coherence, we introduce a form of controlled randomness by applying a *fixed* random permutation $\pi : [D] \mapsto [D]$ to the mask, decoupling the selection of salient features from their fixed positions. This strategically redirects updates away from the most influential features and onto less critical ones. Semantic coherence is maintained—since similar prompts still yield overlapping top-k indices under the fixed permutation—while diversifying the actual features activated in the residual memory. Due to the information redundancy in the LLM activations and the large number of active indices $k$, the application of the random permutation $\pi$ strikes a good balance between diversity and information retention. We ablate its importance in Table 12 of the appendix, showing that both the top-k selection and further permutation are crucial to the performance.

Overall, the end product of TopHash, illustrated in Figure 3, will be a sparse mask $\mathcal{M}(\boldsymbol{a}) \in \mathbb{R}^D$ formally defined as:

$$\mathcal{M}(\boldsymbol{a}) = \pi(\mathcal{T}(\boldsymbol{a})) \text{ for } \mathcal{T}(\boldsymbol{a})_j = \begin{cases} 1 & \text{if } a_j \geq a_{(k)} \ , \\ 0 & \text{otherwise.} \end{cases} \tag{3}$$

### 3.2 Inference: activating knowledge conditional to inference prompt

After editing, the model stores the updated knowledge in the memory module $\boldsymbol{W}_\mathrm{m}$, while the pre-trained knowledge is preserved in the original layer $\boldsymbol{W}_0$. To correctly respond to diverse prompts at inference stage, the model needs to retrieve the relevant knowledge from its memory. To do so, we propose the following two-step process (i) identifying the type of prompt during inference—whether it is an edited prompt, a rephrased version of an edited prompt, or a prompt irrelevant to the edits; and (ii) activating only the relevant knowledge in the memory. Specifically, it should retrieve the relevant knowledge distributed in corresponding combination of columns in $\boldsymbol{W}_\mathrm{m}$ for the edited prompts and their rephrased variants, while deactivating the entire residual memory for irrelevant prompts for preserving locality.

During editing, we construct a database of sparse masks of each edited sample. Let $\boldsymbol{x} \in \mathcal{D}_\mathrm{edit}$ be an edited prompt. Given that the layers prior to the introduced memory module are frozen, the application of the TopHash mechanism will yield the same sparse mask

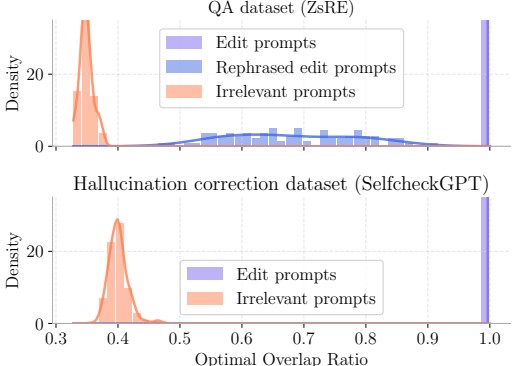

Figure 4: Overlap ratio distributions of active indices between each inference sample and its best match using `LLaMA-3-8B-Instruct`. The y axis is truncated for visualization.

$\mathcal{M}(\boldsymbol{a}(\boldsymbol{x}))$ during both editing and inference, allowing the reliable identification of the prompt as part of the edit set by finding the exact match in the above database. Since transformers cluster semantically similar input prompts together due to their ability to generalize across paraphrases [32], we expect that $\boldsymbol{x}$ should have activations, as well as TopHash masks, with high similarity w.r.t. its semantically rephrased variants. Conversely, semantically unrelated prompts will exhibit low similarity in both activation and mask space to the edited prompt $\boldsymbol{x}$.

We can compare the semantic similarity among prompts via the Hamming distance, denoted as $d_H$, between their respective masks [31]. By populating the database during editing, we can identify the semantically closest edited prompt to the prompt $\boldsymbol{x}$:

$$\boldsymbol{x}_{\text{match}} = \underset{\boldsymbol{x}' \in \mathcal{D}_{\text{edit}}}{\arg\min} \, d_{\text{H}}\big(\mathcal{M}(\boldsymbol{a}(\boldsymbol{x})), \, \mathcal{M}(\boldsymbol{a}(\boldsymbol{x}')))$$

We define the optimal overlapping ratio[3] between a prompt $\boldsymbol{x}$ and its closest edit $\boldsymbol{x}_{\text{match}}$ as:

$$R_{\text{match}}(\boldsymbol{x}) = \frac{1}{N} \left\| \mathcal{M}\big(\boldsymbol{a}(\boldsymbol{x})\big) \wedge \mathcal{M}\big(\boldsymbol{a}(\boldsymbol{x}_{\text{match}}))\right\|_1 \in [0, 1]$$

The overlapping ratio $R_{\text{match}}$ measures the semantic relevance between a prompt and its closest edited prompt, enabling us to distinguish between different prompt types. Figure 4 presents the distribution of $R_{\text{match}}$ for edited prompts, rephrased variants, and irrelevant prompts across two benchmarks. We observe that these prompt categories exhibit distinct modes within the distribution, demonstrating the effectiveness of the proposed TopHash mechanism as a Locality-Sensitive Hashing (LSH) function [30] that clusters semantically similar input prompts.

**Activating the relevant knowledge for the prompt** We intuitively classify the prompt type of the input $\boldsymbol{x}$ based on $R_{\text{match}}$ and a threshold $\tau$ and activate only the relevant knowledge from memory accordingly. Formally, the edited feedforward output at inference is then computed as:

$$\text{FFN}_{\text{edited}}(\boldsymbol{a}(\boldsymbol{x})) = \begin{cases} \boldsymbol{W}_0 \, \boldsymbol{a}(\boldsymbol{x}) + \boldsymbol{W}_{\text{m}} \left(\mathcal{M}(\boldsymbol{a}(\boldsymbol{x}_{\text{match}})) \odot \boldsymbol{a}(\boldsymbol{x})\right) & \text{if } R_{\text{match}}(\boldsymbol{x}) \in [\tau, 1.0], \\ \boldsymbol{W}_0 \, \boldsymbol{a}(\boldsymbol{x}) & \text{if } R_{\text{match}}(\boldsymbol{x}) \in [0, \tau) \end{cases} \quad (4)$$

Specifically, if $R_{\text{match}}(\boldsymbol{x}) \geq \tau$, we identify $\boldsymbol{x}$ as semantically relevant to an edited prompt $\boldsymbol{x}_{\text{match}}$, and apply the mask $\mathcal{M}(\boldsymbol{a}(\boldsymbol{x}_{\text{match}}))$ to its activations to activate the columns in $\boldsymbol{W}_{\text{m}}$ storing the relevant knowledge. This mechanism ensures that semantically rephrased prompts are aligned with their original edited counterparts in representation space, enabling precise retrieval of the relevant residual memory. As a result, it significantly improves generalization to paraphrased queries. Meanwhile, the residual memory is deactivated for prompts considered irrelevant to all edited prompts, i.e., $R_{\text{match}}(\boldsymbol{x}) < \tau$, effectively preserving locality by relying solely on the pre-trained knowledge.

Prior approaches often require access to large corpora of irrelevant samples to enhance locality; either by sampling a different irrelevant sample during each edit [9] or by performing a forward pass on a corpora of up to 100,000 samples to regularize editing [14, 15]. MEMOIR, on the other hand, eliminates the need for such large-scale irrelevant samples during editing and offers a more flexible alternative, as it relies on the introduced conditional knowledge activation mechanism in Equation 4 to decide whether to use the edited or pre-trained knowledge.

## 4 Experiments

In this section, we present a quantitative evaluation of MEMOIR across different benchmarks, LLMs, and numbers of edits, ranging from 1 to 15,000. The best average performance metric is in **bold**.

### 4.1 Experimental setup

**Baselines** We compare MEMOIR against several established baselines. We first consider non-parametric methods, namely GRACE [8], which stores knowledge in an external codebook. We also evaluate parametric methods, including DEFER [8], a method inspired by SERAC [12] for

---

[3]Note that the overlapping ratio is equivalent to the Hamming distance for two binary vectors with the exact same number of zeros and ones.

Table 1: Q&A task results on the ZsRE dataset. $T$ denotes the number of edits. Higher is better for all metrics. The best average performance of reliability, generalization and locality is marked in **bold**.

| Method | $T = 1$ | | | | $T = 10$ | | | | $T = 100$ | | | | $T = 1,000$ | | | |
|---|---|---|---|---|---|---|---|---|---|---|---|---|---|---|---|---|
| | Rel. | Gen. | Loc. | Avg. | Rel. | Gen. | Loc. | Avg. | Rel. | Gen. | Loc. | Avg. | Rel. | Gen. | Loc. | Avg. |
| | LLaMA-3-8B | | | | | | | | | | | | | | | |
| FT | 0.49 | 0.49 | 0.65 | 0.54 | 0.18 | 0.15 | 0.12 | 0.15 | 0.13 | 0.11 | 0.02 | 0.09 | 0.13 | 0.12 | 0.01 | 0.09 |
| ROME [11] | 0.99 | 0.96 | 0.96 | 0.97 | 0.61 | 0.59 | 0.41 | 0.54 | 0.08 | 0.08 | 0.02 | 0.06 | 0.03 | 0.03 | 0.02 | 0.03 |
| MEMIT [14] | 0.99 | 0.97 | 0.98 | 0.98 | 0.68 | 0.67 | 0.73 | 0.69 | 0.03 | 0.03 | 0.01 | 0.02 | 0.00 | 0.00 | 0.00 | 0.00 |
| DEFER [8] | 0.82 | 0.86 | 0.68 | 0.79 | 0.66 | 0.67 | 0.45 | 0.59 | 0.33 | 0.32 | 1.00 | 0.55 | 0.31 | 0.31 | 1.00 | 0.54 |
| GRACE [8] | 1.00 | 0.46 | 1.00 | 0.82 | 1.00 | 0.42 | 1.00 | 0.81 | 1.00 | 0.39 | 1.00 | 0.80 | 1.00 | 0.37 | 1.00 | 0.79 |
| WISE [9] | 0.92 | 0.84 | 1.00 | 0.92 | 0.78 | 0.74 | 0.98 | 0.83 | 0.62 | 0.60 | 1.00 | 0.74 | 0.66 | 0.64 | 1.00 | 0.77 |
| AlphaEdit [15] | 0.98 | 0.89 | 1.00 | 0.96 | 0.93 | 0.85 | 0.98 | 0.92 | 0.91 | 0.79 | 0.94 | 0.88 | 0.84 | 0.77 | 0.56 | 0.72 |
| **MEMOIR (Ours)** | 1.00 | 1.00 | 1.00 | **1.00** | 0.97 | 0.89 | 1.00 | **0.95** | 0.96 | 0.89 | 1.00 | **0.95** | 0.94 | 0.85 | 1.00 | **0.93** |
| | Mistral-7B | | | | | | | | | | | | | | | |
| FT | 0.58 | 0.54 | 0.91 | 0.68 | 0.39 | 0.39 | 0.50 | 0.43 | 0.11 | 0.10 | 0.02 | 0.08 | 0.16 | 0.13 | 0.01 | 0.10 |
| ROME [11] | 0.79 | 0.77 | 0.98 | 0.85 | 0.58 | 0.57 | 0.75 | 0.63 | 0.05 | 0.05 | 0.02 | 0.04 | 0.04 | 0.04 | 0.02 | 0.03 |
| MEMIT [14] | 0.81 | 0.79 | 0.99 | 0.86 | 0.46 | 0.45 | 0.61 | 0.51 | 0.00 | 0.00 | 0.01 | 0.00 | 0.04 | 0.04 | 0.02 | 0.03 |
| DEFER [8] | 0.59 | 0.68 | 0.79 | 0.69 | 0.44 | 0.43 | 1.00 | 0.62 | 0.40 | 0.40 | 1.00 | 0.60 | 0.40 | 0.39 | 1.00 | 0.60 |
| GRACE [8] | 1.00 | 0.36 | 1.00 | 0.79 | 1.00 | 0.15 | 1.00 | 0.72 | 1.00 | 0.15 | 1.00 | 0.72 | 1.00 | 0.02 | 1.00 | 0.67 |
| WISE [9] | 0.98 | 0.97 | 1.00 | 0.98 | 0.92 | 0.89 | 1.00 | 0.94 | 0.87 | 0.80 | 1.00 | 0.89 | 0.70 | 0.67 | 1.00 | 0.79 |
| AlphaEdit [15] | 0.83 | 0.77 | 1.00 | 0.87 | 0.87 | 0.75 | 0.99 | 0.87 | 0.86 | 0.74 | 0.95 | 0.85 | 0.85 | 0.72 | 0.68 | 0.75 |
| **MEMOIR (Ours)** | 1.00 | 0.99 | 1.00 | **1.00** | 0.97 | 0.94 | 1.00 | **0.97** | 0.95 | 0.91 | 1.00 | **0.95** | 0.94 | 0.89 | 1.00 | **0.94** |

Table 2: Hallucination correction task results on the SelfCheckGPT dataset. $T$ denotes the number of edits. The reliability metric refers to perplexity, where *smaller* value indicates better performance. Due to large value ranges, we use scientific notation (e.g., 8.05e1 denotes 80.5) for clarity.

| Method | LLaMA-3-8B | | | | | | | | Mistral-7B | | | | | | | |
|---|---|---|---|---|---|---|---|---|---|---|---|---|---|---|---|---|
| | $T = 1$ | | $T = 10$ | | $T = 100$ | | $T = 600$ | | $T = 1$ | | $T = 10$ | | $T = 100$ | | $T = 600$ | |
| | Rel.(↓) | Loc.(↑) | Rel. | Loc. | Rel. | Loc. | Rel. | Loc. | Rel. | Loc. | Rel. | Loc. | Rel. | Loc. | Rel. | Loc. |
| FT | 8.24 | 0.85 | 8.05e1 | 0.37 | 5.72e2 | 0.13 | 2.36e5 | 0.05 | 2.50e1 | 0.38 | 1.00e2 | 0.03 | 1.59e3 | 0.00 | – | – |
| ROME [11] | 1.52 | 0.97 | 2.02e1 | 0.61 | 1.26e5 | 0.02 | 4.07e4 | 0.02 | 2.04 | 0.99 | 3.45 | 0.92 | 1.04e2 | 0.03 | 2.41e2 | 0.01 |
| MEMIT [14] | 1.30 | 0.99 | 4.05e2 | 0.93 | 4.56e10 | 0.01 | 2.95e6 | 0.00 | 1.64 | 1.00 | 1.59e1 | 0.89 | 9.72e1 | 0.04 | 1.32e2 | 0.02 |
| DEFER [8] | 5.69e2 | 0.83 | 2.36e2 | 0.82 | 7.02e1 | 1.00 | 6.85e1 | 1.00 | 1.46e1 | 0.94 | 3.22e1 | 0.99 | 3.22e1 | 0.99 | 3.22e1 | 1.00 |
| GRACE [8] | 1.05 | 1.00 | 7.10e1 | 1.00 | 7.12e1 | 1.00 | 7.73e1 | 1.00 | 1.39 | 1.00 | 5.97 | 1.00 | 9.53 | 1.00 | 9.57 | 1.00 |
| WISE [9] | 4.93e1 | 0.98 | 1.46 | 0.95 | 2.10 | 0.99 | 3.20 | 0.99 | 1.40 | 1.00 | 2.56 | 0.94 | 1.31 | 0.99 | 5.21 | 0.93 |
| AlphaEdit [15] | 1.58 | 1.00 | 3.12 | 0.98 | 5.97 | 0.93 | 8.49e1 | 0.05 | 1.75 | 1.00 | 1.76 | 1.00 | 2.87 | 0.98 | 1.70e2 | 0.88 |
| **MEMOIR (Ours)** | 1.00 | 1.00 | 1.01 | 1.00 | 1.07 | 1.00 | 1.25 | 1.00 | 1.00 | 1.00 | 1.02 | 1.00 | 1.09 | 1.00 | 1.22 | 1.00 |

inference-time routing; methods based on causal tracing, including ROME [11], MEMIT [14], and ALPHAEDIT [15]; and memory-based approaches such as WISE [9]. Finally, we include direct fine-tuning (FT) as an additional baseline. Detailed descriptions and implementation details for all methods are provided in Appendix A.

**Models** We perform our experiments on four different widely used autoregressive LLMs: LLaMA-3-8B-Instruct [16], Mistral-7B[18], LLaMA-2-7B[19] (results provided in Appendix B.1) and GPT-J-6B [20]. For brevity, we refer to them as LLaMA-3, Mistral, LLaMA-2, and GPT-J throughout this section.

**Evaluation metrics** We follow prior work and evaluate the effectiveness of different editing methods based on three metrics [9]: *reliability*, *generalization* and *localization*, which are respectively computed by model's performance on edit, rephrased edit and irrelevant samples.

## 4.2 Main Experimental Results

**Question & Answering** Table 1 presents the results on the ZsRE [17] dataset for the question answering task, evaluated across a range of total edits from 1 to 1000. ZsRE is a context-free Q&A benchmark designed to assess a model's ability to store and retrieve specific factual knowledge. We observe that MEMOIR consistently achieves the most balanced performance across all evaluation settings. On LLaMA-3, MEMOIR maintains a reliability of 0.94 and a generalization of 0.85, with its locality reaching 1.0 across different edit counts. In contrast, other parametric methods exhibit substantial degradation as the number of edits increases. For example, WISE drops from an average metric of 0.92 to 0.77 as the number of edits grows from 1 to 1,000. A similar trend is observed with AlphaEdit, which declines from 0.96 to 0.72. Although GRACE, a non-parametric baseline,

maintains high reliability, it suffers from poor generalization, leading to a significantly lower average metric overall compared to `MEMOIR`. In summary, `MEMOIR` offers the best trade-off between reliability, generalization, and locality, achieving an average metric of 0.93 on LLaMA-3 with 1,000 total edits—outperforming all prior methods by a substantial margin of 0.14. Similar results are observed with Mistral, where `MEMOIR` again delivers the highest average metric, demonstrating its robustness and effectiveness across different LLMs.

**Hallucination correction**   LLMs are prone to hallucinations—generating fluent but factually inaccurate or misleading content—which poses a significant challenge for their deployment in domains that demand high factual precision. We validate here the effectiveness of `MEMOIR` on correcting hallucinations on the SelfCheckGPT [33] dataset, which consists of Wikipedia-style biographies generated by GPT-3 [1]. The dataset is annotated for factual hallucinations, with ground-truth replacements verified against Wikipedia. We note that we report only *locality* and *reliability* using perplexity (PPL) metric, following [9], as no suitable metric for generalization exists in this setting. As shown in Table 2, `MEMOIR` again achieves the best balanced performance, with its advantages becoming even more significant with a larger number of edits. Under the most challenging setting of 600 edits, `MEMOIR` maintains a saturated locality while delivering a perplexity metric, 61% and 77% smaller than WISE, the second-best performing approach, respectively on LLaMA-3 and on Mistral. These results further confirm `MEMOIR`'s robustness in sustaining model quality during long editing sequences.

**Out-of-distribution (OOD) generalization**   Effective model editing should generalize from structured prompts to distributionally shifted inputs, where the shift reflects increased complexity rather than domain change [34, 35]. We evaluate the performance of `MEMOIR` on OOD settings using the Temporal dataset [36], which consists of a

Table 3: OOD generalization on Temporal dataset with `GPT-J-6B` model comparing `MEMOIR` with prior works.

| Method | $T = 10$ | | | | $T = 75$ | | | |
|---|---|---|---|---|---|---|---|---|
| | Rel. | OOD Gen. | Loc. | Avg. | Rel. | OOD Gen. | Loc. | Avg. |
| *w/o Editing* | 0.56 | 0.21 | — | — | 0.56 | 0.21 | — | — |
| MEMIT [14] | 0.88 | 0.28 | 0.98 | 0.71 | 0.41 | 0.18 | 0.14 | 0.24 |
| GRACE [8] | 0.97 | 0.28 | 1.00 | 0.75 | 0.97 | 0.28 | 1.00 | 0.75 |
| WISE [9] | 0.99 | 0.36 | 0.98 | 0.78 | 0.96 | 0.37 | 1.00 | 0.78 |
| AlphaEdit [15] | 0.89 | 0.27 | 0.99 | 0.72 | 0.85 | 0.27 | 0.96 | 0.69 |
| **MEMOIR (Ours)** | 0.99 | 0.37 | 1.00 | **0.79** | 0.99 | 0.38 | 1.00 | **0.79** |

prefix paired with a Wikipedia description and a model-generated description. As shown in Table 3, `MEMOIR` outperforms the state-of-the-art consistently at both $T = 10$ and $T = 75$, indicating strong adaptation to novel knowledge and effective generalization.

**Multi-hop reasoning**   In appendix B.5 we assess the efficacy of `MEMOIR` in multi-hop reasoning, a more challenging scenario. We evaluate on the RippleEdits benchmark [37], where `MEMOIR` demonstrates better multi-hop reasoning generalization than all baseline methods.

## 4.3   Scaling with longer sequences of edits

The ultimate goal of lifelong model editing algorithm is to continually deliver reliable performance with an increasingly large number of edits without performance degradation on previously edited knowledge. Here we extend the total number of edits to 7,000 and 15,000 edits on LLaMA-3, almost exploiting all samples from ZsRE dataset. Table 4 presents the performance of `MEMOIR`

Table 4: Editing performance on LLaMA-3 with 7,000 and 15,000 edits.

| Method | $T = 7,000$ | | | | $T = 15,000$ | | | |
|---|---|---|---|---|---|---|---|---|
| | Rel. | Gen. | Loc. | Avg. | Rel. | Gen. | Loc. | Avg. |
| MEMIT [14] | 0.00 | 0.00 | 0.00 | 0.00 | – | – | – | – |
| GRACE [8] | 1.00 | 0.31 | 1.00 | 0.77 | 0.99 | 0.28 | 1.00 | 0.76 |
| WISE [9] | 0.59 | 0.57 | 1.00 | 0.72 | 0.43 | 0.41 | 1.00 | 0.61 |
| AlphaEdit [15] | 0.02 | 0.02 | 0.00 | 0.01 | 0.02 | 0.02 | 0.00 | 0.01 |
| **MEMOIR (Ours)** | 0.92 | 0.85 | 0.99 | **0.92** | 0.87 | 0.78 | 0.98 | **0.88** |

using LLaMA-3 model and shows `MEMOIR`'s remarkable scalability to the total number of edits beyond prior baselines. Averaged across all metrics, `MEMOIR` remains by far the best performing method on both 7,000 and 15,000 edits, with a margin of over 0.12 compared to prior methods. A full performance trajectory from 1,000 to 3,000 edits, comparing `MEMOIR` against key baselines on both LLaMA-3 and Mistral is provided in Figure 5 in Appendix.

## 5   Analysis and Discussion

We provide additional analysis of `MEMOIR`'s performance. Unless noted otherwise, experiments are based on the LLaMA-3 model with 1,000 sequential edits on ZsRE.

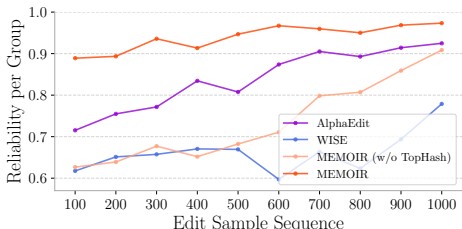
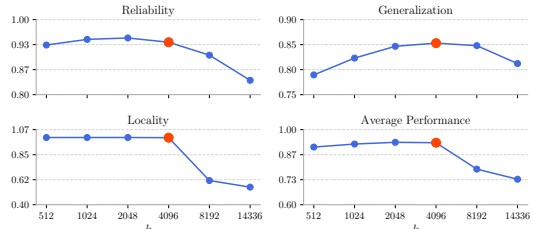

Figure 5: Reliability on sequentially edited samples (grouped per 100 edits).

Figure 6: Performance vs. number of active indices $k$ in TopHash.

**Ablation on conditional knowledge activation**    We evaluate the effectiveness of inference-time conditional knowledge activation strategy introduced in Section 3.2. Specifically, we directly use the mask based on the input activations without the matching step of Equation 4 that enables either the activation of only relevant memory knowledge or the activation of the memory module altogether. Table 5 shows that disabling conditional knowledge activation results in a noticeable drop in generalization performance, since the model is unable to precisely activate the parameters that store relevant knowledge. Moreover, the absence of conditional activation significantly degrades locality: performance on irrelevant prompts drops below 0.7, whereas MEMOIR maintains full performance on these samples by relying solely on pre-trained knowledge.

Table 5: Ablation for the performance of MEMOIR without conditional knowledge activation (abbreviated as K.A.).

| Metric | Rel. | Gen. | Loc. | Avg. |
|---|---|---|---|---|
| MEMOIR | 0.94 | 0.85 | 1.00 | **0.93** |
| MEMOIR w/o K.A. | 0.94 | 0.77 | 0.68 | 0.80 |

**Reduced forgetting during editing**    MEMOIR distributes knowledge updates across distinct parameter subsets to reduce interference and mitigate forgetting of previously edited samples. To quantify this effect, we report the post-edit accuracy for each sample in the editing sequence. As shown in Figure 5, MEMOIR exhibits the smallest degradation in accuracy for earlier edits. Specifically, it maintains an average accuracy of near 0.90 for the first 100 edits, even after all 1000 edits have been applied. In contrast, the accuracy of the second-best method, AlphaEdit, on the first 100 edits drops to 0.71. This pattern holds across the full sequence, highlighting MEMOIR's ability to preserve earlier edits, while baseline methods experience significant forgetting.

**Impact of number of active indices $k$**    We ablate the number of active indices $k$ used in TopHash in Figure 6. The maximum value, 14336, corresponds to the total number of residual memory columns in LLaMA-3. Fewer selected features limit the model's ability to capture edits due to fewer parameters, reducing accuracy and generalization, while high $k$ degrades performance by causing excessive parameter overwriting. For locality, a low number of active indices degrades the quality of the captured semantic relevance by the masks to trigger routing, while a high number causes substantial mask overlap, making edits' fingerprints indistinguishable. We adopt 4096 groups for both LLaMA-3 and Mistral as a balanced choice.

## 6    Related Works

**Sparse neural activations**    Sparsity and non-overlapping activations have been widely explored in continual learning to mitigate catastrophic forgetting by localizing updates and reducing interference. Early works [38, 39] introduced this idea, later extended by methods like PackNet [23] and Supermasks-in-Superposition [24], which allocate disjoint parameter subsets per task but require task identities at inference and assume clear task boundaries. Gradient-based methods such as GPM [26] and SPARCL [27] improve parameter efficiency via orthogonal updates or activation sparsity. Even standard Dropout [40] implicitly encourages non-overlapping activations [25], while heterogeneous dropout across tasks aids activation sparsity [28]. However, these approaches remain confined to continual learning, often relying on irrelevant data or explicit regularization. In contrast, we tackle lifelong model editing without task supervision or boundaries, inducing non-overlapping updates by directly sparsifying input activations to a residual memory, enabling scalable, localized knowledge editing.

**Model editing on LLMs**  Prior work in model editing falls into two categories: parametric and non-parametric methods. Parametric methods typically modify model weights through meta-learning, locating-then-editing strategies, or auxiliary memory modules. Meta-learning approaches like KE [41] and MEND [42] use hypernetworks to incorporate updates while minimizing disruption to previous knowledge. Locating-then-editing methods such as ROME [11] and MEMIT [14] identify and edit specific modules where facts are stored, while AlphaEdit [15] improves them by projecting edits onto the null space of preserved knowledge to avoid corrupting previous information. Auxiliary module methods, including SERAC [12] and T-Patcher [13], extend the model with lightweight components for storing and routing updates. WISE [9] further advances this line by introducing routing mechanisms and memory sharding to localize edits and mitigate cross-edit interference. Despite their effectiveness, parametric methods often suffer from forgetting over long edit sequences. Non-parametric methods store knowledge externally to preserve original weights. GRACE [8] retrieves discrete codebook entries to overwrite intermediate activations during inference to enable precise and local edits, while LOKA [10] tackles simultaneous editing and unlearning by storing updates in task-specific or shared memory. However, these methods rely on exact input matches, limiting generalization to paraphrased queries.

## 7   Conclusion and Limitations

We introduced `MEMOIR`, a scalable framework for lifelong model editing that balances reliability, generalization, and locality. By sparsifying input activations through sample-dependent masks and confining updates to a dedicated parameter space, `MEMOIR` enables precise knowledge injection without disrupting the pre-trained model's capabilities. At inference, it retrieves relevant updates by comparing sparse activation patterns, allowing edits to generalize to rephrased queries while preserving the model's behavior on unrelated prompts. Extensive experiments show that `MEMOIR` outperforms state-of-the-art methods across question answering, hallucination correction, and out-of-distribution generalization tasks, scaling to thousands of edits with minimal forgetting. Overall, `MEMOIR` offers a practical tool for improving the factual accuracy and safety of deployed language models.

Despite these advances, `MEMOIR` has certain limitations. While it scales better than prior methods, it modifies only a single linear layer, which may limit its ability to handle long-horizon edits or knowledge requiring broader model changes. Extending the approach to multiple layers or hierarchical editing remains a promising direction. Finally, our experiments focus on decoder-only transformers; applying `MEMOIR` to multi-modal or encoder-decoder models is left for future work.

## Acknowledgments

The authors thank Guillermo Ortiz-Jimenez for constructive discussions and comments.

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

# Appendix Overview

In the Appendix, we provide additional details organized as follows:

# A  Experimental Setup

## A.1  Datasets

Table 6: Example from the ZsRE dataset (Q&A task).

| ZsRE dataset | |
|---|---|
| Prompt $x_e$ | What network aired *Faszination Wissen*? |
| Rephrased prompt $x'_e$ | Which station aired *Faszination Wissen*? |
| Edit target $y_e$ | SVT1 |
| Irrelevant prompt $x_{irr}$ | When does english dragon ball super come out |
| Irrelevant target $y_{irr}$ | starting on January 7, 2017 |

Table 7: Example from the SelfCheckGPT dataset (hallucination correction task).

| SelfCheckGPT dataset | |
|---|---|
| Prompt $x_e$ | This is a Wikipedia passage about bill brown goalkeeper. Bill Brown (born 28 April 1932) is a former Scottish football goalkeeper. He is best known for his time at Celtic, where he made over 500 appearances in all competitions between 1951 and 1967. |
| Edit target $y_e$ | He started his senior career with Dundee as a teenager and made over 200 appearances in the Scottish Football League. |
| Irrelevant prompt $x_{irr}$ | NFL commissioner Roger Goodell has stated that for the 2017 season all replay reviews will be centralized in the league's gameday command center and d |
| Irrelevant target $y_{irr}$ | ecided by the senior vice president of officiating. Goodell's comments came from an interview on Mike & Mike on ESPN hours before the Competition Comm |

## A.2  Metrics

The metrics for Q&A and hallucination correction are introduced in the main paper Sec. 4.1. Here, we detail the implementation specific to the Temporal dataset. We adopt the standard evaluation protocol based on reliability on Temporal dataset as performed for both Q&A and hallucination correction tasks.

In the Temporal dataset, each instance is constructed as follows: the prefix $x_e$ is extracted from the first paragraph of an entity's Wikipedia page, and a noisy paragraph $y_e$ generated by GPT-4 [43] about the same emerging entity is sampled. Although $y_e$ may contain inaccuracies, it often provides useful cues. These $(x_e, y_e)$ pairs are presented as editing prompts to Model Editors. To assess out-of-distribution (OOD) generalization in naturally complex contexts, the dataset provides $y_{ood}$, which is sampled from the suffix portion of the same Wikipedia entry. This design allows us to evaluate how well Model Editors generalize beyond the edited prompt, within a realistic and entity-consistent context.

Table 8: Example from the Temporal dataset (OOD task).

| Temporal dataset | |
| --- | --- |
| Prompt $x_e$ | Adam Silver |
| Edit target $y_e$ | is an American lawyer and the current commissioner of the National Basketball Association (NBA). He has been praised for his progressive leadership, focusing on both the on-court game and the business side of the league. Under Silver's tenure, the NBA has experienced significant growth and expansion, implementing various rule changes and advancements in technology to enhance the fan experience. Silver is also known for his commitment to social justice and player empowerment, encouraging players to use their platforms to speak out on important societal issues. His forward-thinking approach has helped solidify the NBA's position as a global sports powerhouse. |
| OOD prompt and target $x_e, y_{ood}$ | Adam Silver, the current commissioner of the National Basketball Association (NBA), has made a significant impact on the league since taking the position in 2014. Prior to becoming the commissioner, Silver had already established a solid foundation within the NBA, holding various roles including chief operating officer and deputy commissioner. Under his leadership, the NBA has experienced considerable growth, both economically and globally. One particular milestone during Silver's tenure was his handling of the Donald Sterling controversy in 2014, where he forced Sterling to sell the Los Angeles Clippers and banned him from all NBA games and events. Silver's decisive action showcased his commitment to maintaining the integrity of the NBA and addressing issues of racism within the league. |
| Irrelevant prompt $x_{irr}$ | Written by Gerald H. Nelson ž2022 Fred B. Burch Album This song officially appears on the Red Rose Speedway (2018) Official album. Sessions This song has |
| Irrelevant target $y_{irr}$ | been recorded during the following sessions Red Rose Speedway sessions at Olympic Studios (March 1972)Other unreleased songs from the double-album Re |

Following prior work [9, 11], out-of-scope data $x_{loc}$ is drawn from the Pile [44], the pretraining corpus for GPT-J-6B. Consistent with [9], we evaluate using a *probability threshold* criterion: editing is considered successful if the loss $L_\theta^T(x_e, y_{ood})$ falls below $\delta = -\log(0.8)$. This reflects the assumption that multiple plausible continuations are acceptable. The OOD generalization metric is thus defined as:

$$\text{OOD Gen.} = \frac{1}{T} \sum_{t=1}^{T} \mathbb{1}\{L_\theta^T(x_e^t, y_{ood}^t) < \delta\},$$

where $T$ is the number of test instances. This formulation quantifies the fraction of successful generalizations under the threshold criterion.

### A.3  Baselines

**ROME** [11] identified the feedforward MLPs in the middle layers store the factual knowledge by tracing the causal effects of hidden state activations based on causal mediation analysis. Furthermore, they introduced a Rank-One Model Editing method to alter the parameters storing factual knowledge and modify the model's behavior.

**MEMIT** [14] extends ROME to batch edits and multiple layers. It first identified critical MLP layers, then each layer receives an update expectedly calculated to insert new memories. This enables MEMIT to scale to thousands of edits.

**DEFER** [8] is a re-implementation of SERAC [12] in [8] which avoids changing the weights of the original model by introducing two auxiliary modules: a classifier $g$ and a one-layer network $o$. During editing, $g$ and $o$ are fine-tuned jointly. At the inference stage, the classifier $g$ routes between the predictions of the LLMs or the predictions by the one-layer network $o$.

**GRACE** [8] uses a non-parametric codebook to store edited knowledge. Each edit either adds, expands or splits a key-value pair to the codebook. At inference, it retrieves value with the closest key and overwrites the output activations of an intermediate layer. GRACE supports thousands of edits without altering model weights, enabling precise and localized edits.

**WISE** [9] maintains two memory paths: the original memory (long-term memory) and a side memory (working memory) updated with all edits. Edits are applied only to the side memory, and a router decides which memory to use per prompt during inference. WISE partitions updates across two subspaces to prevent interference, merging them into the side memory before inference.

**AlphaEdit** [15] adds a projection step to standard locate-then-edit methods [11, 14] to prevent interference with previous knowledge. It projects the perturbation $\Delta$ into the null space of edited knowledge, ensuring outputs of previous edits remain minimally affected after new updates.

## A.4  Experimental Details

### A.4.1  Training Details

All experiments are conducted on a single NVIDIA A100 GPU. Editing 1,000 Q&A samples with MEMOIR takes approximately 3 hours for LLaMA-3 and 4 hours for Mistral, using an SGD optimizer. For the hallucination dataset, editing 600 samples requires around 4 hours on LLaMA-3 and 6 hours on Mistral.

For all our experiments, we follow the lifelong model editing setting [8, 9] and perform edits on singular edits, i.e., the batch size during editing is always 1, such that the model is edited immediately upon the arrival of new edits to correct model behavior in a timely manner.

We reproduced the baseline results using the EasyEdit repository [45], a popular repository involving the implementations for multiple knowledge editing methods. For the results of baseline methods including FT, ROME, MEMIT, GRACE and WISE in ZsRE and SelfcheckGPT dataset on LLaMA-2 and Mistral, we report the evaluated results from [9], and we follow their evaluation settings in our reproduction of these baselines. For all other results, we detail the setting for each method below:

### A.4.2  Baseline implementations

**ROME**  For ROME, we follow the default setting in their sourcecode on GPT-J, as well as the implementation in [14] and [9], to edit the 5th layer in the LLM for both LLaMA-3 and Mistral. The covariance statistics are computed using 100,000 samples from Wikitext in fp32 precision.

**MEMIT**  For both LLaMA and Mistral models, MEMIT updates layers [4, 5, 6, 7, 8] and sets $\lambda$, the covariance adjustment factor, to 15,000, following the settings in [9]. Following ROME [11], it uses 100,000 Wikipedia samples.

**DEFER**  For DEFER, we follow the hyper-parameter settings in [9], using a learning rate of 7e-5, number of training steps of 100, and threshold of 0.5.

**GRACE**  Following the setup in [8], we use a learning rate of 1.0 and adopt the replace_last strategy, which replaces only the activation of the final token in autoregressive settings. For the deferral radius $\epsilon$, we evaluate values of 1.0, 3.0, and 10.0, reporting the best-performing result.

**FT**  For the FT baseline, we use the Adam optimizer with a learning rate of 5e-4 and train for 50 steps per edit.

**WISE**  For WISE, we adopt the hyperparameter settings provided in [9] for both LLaMA-2 and Mistral. Specifically, we use the following configuration: optimizer is SGD with learning rate $\eta = 1.0$,

mask ratio $\rho = 0.2$, $\alpha = 5.0$, $\beta = 20.0$, $\gamma = 10.0$, merge weights $\lambda = 0.5$, training steps to be 70, and number of knowledge shards $k = 2$. For LLaMA-3, we use the same hyperparameters as for LLaMA-2, but reduce the number of training steps to 30 to improve computational efficiency.

**AlphaEdit** For AlphaEdit, we adopt the same hyperparameter settings as in [15] to ensure fair comparison. Specifically, for the LLaMA-3 (8B) model, we edit critical layers [4, 5, 6, 7, 8]. To compute hidden representations at each critical layer, we perform 25 optimization steps with a learning rate of 0.1. For LLaMA-2 and Mistral, we apply the same configuration. For the GPT-J model, we target layers [3, 4, 5, 6, 7, 8], and similarly perform 25 optimization steps per layer with a learning rate of 0.5. As in MEMIT [14], it uses $\lambda = 15,000$ and 100,000 Wikipedia samples.

### A.4.3 Implementation details for `MEMOIR`

For all model architectures, including LLaMA-2, LLaMA-3, Mistral, and GPT-J, we edit the layer `model.layers[27].mlp.down_proj.weight` for LLaMA-2, LLaMA-3 and Mistral, and `transformer.h[21].mlp.fc_out.weight` for GPT-J. We use the SGD optimizer with a learning rate of 1.0 and a gradient clipping with clipping threshold of 1.0. The number of iterations per edit is set as 70 for LLaMA-2, Mistral, and GPT-J, and reduces to 30 for LLaMA-3 for accelerating computation.

The number of active indices $k$ for TopHash is set as 4096 for LLaMA-3, LLaMA-2, Mistral, and GPT-J architecture. The threshold for conditional knowledge activation $\tau$ is set as 0.4 for LLaMA-3 and Mistral, 0.46 for LLaMA-2, and 0.45 for GPT-J.

During editing, we iteratively go through each sample in the edit sequence to update the parameters of the introduced memory module and incorporate new knowledge. We follow [9] to augment each edit sample by generating 10 random token sequences of length 10 with the LLM itself as a prefix to the edit prompt. Note that we do not incorporate any irrelevant samples into the training process for `MEMOIR`, while they are required for several prior baselines [9, 15, 14].

### A.4.4 Experiments for a small number of edits

In Table 1 and Table 2, the total number of edits gradually increases from 1 to 1,000 for the ZsRE dataset and from 1 to 600 for the SelfcheckGPT dataset. To reduce the variance introduced by individual edit samples in experiments involving a small number of edits ($T = 1$ to $T = 100$), we report results averaged over multiple runs using different edit samples. For example, results for $T = 1$ on the ZsRE dataset are averaged over 1,000 independent experiments, each using a different edit sample. Similarly, for $T = 10$ on the SelfcheckGPT dataset, we average results over 60 runs with distinct edit samples. In total, all experiments span 1,000 edit samples on ZsRE and 600 on SelfcheckGPT.

## B   Additional Results

### B.1   Model editing on LLaMA-2-7B

We provide in this section the result for editing LLaMA-2-7B model with `MEMOIR` on both Q&A and hallucination correction tasks. We demonstrate that `MEMOIR` continues to deliver state-of-the-art performance, similarly to the results on LLaMA-3 and Mistral in main text.

**Q&A** We present the results on the ZsRE dataset with LLaMA-2 models in Table 9. The results demonstrate that `MEMOIR` delivers the best performance against the compared baseline approaches, where its advantage becomes increasingly pronounced with larger numbers of total edits. For a total of 1,000 edits, `MEMOIR` achieves an average score of 0.95 across the three evaluation metrics, namely reliability, generalization, and locality, outperforming the second-best method, WISE, by a margin of 0.13. Notably, while most baselines show significant performance degradation as the number of edits increases—particularly in generalization and locality—`MEMOIR` maintains robust performance across all three axes.

**Hallucination correction** We report results on the Hallucination setting using the SelfCheckGPT dataset with LLaMA-2 in Table 10. Notably, while several baselines suffer significant degradation in

Table 9: Editing results for QA setting (ZsRE dataset) on LLaMA-2-7B. $T$ denotes the number of edits.

| Method | $T = 1$ | | | | $T = 10$ | | | | $T = 100$ | | | | $T = 1000$ | | | |
|---|---|---|---|---|---|---|---|---|---|---|---|---|---|---|---|---|
| | Rel. | Gen. | Loc. | Avg. | Rel. | Gen. | Loc. | Avg. | Rel. | Gen. | Loc. | Avg. | Rel. | Gen. | Loc. | Avg. |
| FT | 0.57 | 0.52 | 0.96 | 0.68 | 0.48 | 0.48 | 0.76 | 0.57 | 0.30 | 0.27 | 0.23 | 0.27 | 0.19 | 0.16 | 0.03 | 0.13 |
| ROME [11] | 0.85 | 0.80 | 0.99 | 0.88 | 0.64 | 0.62 | 0.75 | 0.67 | 0.23 | 0.22 | 0.04 | 0.16 | 0.01 | 0.01 | 0.00 | 0.01 |
| MEMIT [14] | 0.84 | 0.81 | 0.99 | 0.88 | 0.58 | 0.58 | 0.85 | 0.67 | 0.02 | 0.02 | 0.02 | 0.02 | 0.04 | 0.04 | 0.02 | 0.03 |
| DEFER [8] | 1.00 | 0.99 | 0.90 | 0.96 | 0.94 | 0.93 | 0.74 | 0.87 | 0.87 | 0.85 | 0.65 | 0.79 | 0.57 | 0.57 | 0.50 | 0.55 |
| GRACE [8] | 0.99 | 0.36 | 1.00 | 0.78 | 0.96 | 0.16 | 1.00 | 0.71 | 0.96 | 0.15 | 1.00 | 0.70 | 0.93 | 0.08 | 1.00 | 0.67 |
| WISE [9] | 0.98 | 0.92 | 1.00 | 0.97 | 0.94 | 0.88 | 1.00 | 0.94 | 0.90 | 0.81 | 1.00 | 0.90 | 0.77 | 0.72 | 1.00 | 0.83 |
| AlphaEdit [15] | 1.00 | 0.92 | 1.00 | 0.97 | 0.95 | 0.81 | 0.99 | 0.92 | 0.94 | 0.89 | 0.94 | 0.92 | 0.86 | 0.73 | 0.53 | 0.71 |
| **MEMOIR (Ours)** | 1.00 | 0.96 | 1.00 | **0.99** | 0.99 | 0.92 | 1.00 | **0.97** | 0.98 | 0.91 | 1.00 | **0.96** | 0.97 | 0.90 | 1.00 | **0.96** |

Table 10: Editing results for Hallucination setting (SelfCheckGPT dataset) on LLaMA-2-7B. $T$ denotes the number of edits.

| Method | $T = 1$ | | $T = 10$ | | $T = 100$ | | $T = 600$ | |
|---|---|---|---|---|---|---|---|---|
| | Rel. (PPL$\downarrow$) | Loc. ($\uparrow$) | Rel. ($\downarrow$) | Loc. ($\uparrow$) | Rel. ($\downarrow$) | Loc. ($\uparrow$) | Rel. ($\downarrow$) | Loc. ($\uparrow$) |
| FT | 4.41 | 0.96 | 1.26e1 | 0.71 | 3.31e1 | 0.41 | 6.92e1 | 0.26 |
| ROME [11] | 1.68 | 0.99 | 2.04 | 0.94 | 9.42e1 | 0.05 | 1.05e2 | 0.02 |
| MEMIT [14] | 1.66 | 1.00 | 2.36 | 0.97 | 7.67e1 | 0.05 | 1.08e2 | 0.02 |
| DEFER [8] | 1.03 | 0.91 | 1.37 | 0.86 | 9.30 | 0.76 | 1.43e1 | 0.63 |
| GRACE [8] | 2.21 | 1.00 | 8.67 | 1.00 | 9.67 | 1.00 | 9.34 | 1.00 |
| WISE [9] | 1.91 | 1.00 | 1.04 | 1.00 | 1.14 | 1.00 | 3.12 | 0.99 |
| AlphaEdit [15] | 1.11 | 1.00 | 1.22 | 0.99 | 3.12 | 0.97 | 3.62e2 | 0.86 |
| **MEMOIR (Ours)** | 1.16 | 1.00 | 1.09 | 1.00 | 1.17 | 1.00 | 1.48 | 0.99 |

reliability as the number of edits increases, e.g., ROME and MEMIT reaching over 100 in perplexity at $T = 600$, MEMOIR maintains low perplexity (1.48) and high locality =(0.99). This indicates that our method successfully incorporates new knowledge without disrupting unrelated predictions. On average across all edit scales and metrics, MEMOIR demonstrates a robust scalability in correcting hallucinations.

## B.2 Activation centering during mask construction

As described in the main text, the mask selection procedure involves a centering step, where a reference activation vector is subtracted from the activations of the input prompt $x$. Specifically, this centering step suppresses features that exhibit consistently high magnitudes across different prompts, allowing the mask to focus on more distinctive features for each prompt. In this section, we describe the computation of the reference vector.

The irrelevant prompts used to compute the reference mean are drawn from the ZsRE dataset, ensuring no overlap with the edited prompts. We perform an ablation study by varying the number of irrelevant prompts used for mean estimation, ranging from 10 to 1,000. We also include the results for not performing such feature centering process.

Results in Table 11 demonstrate that the performance of MEMOIR is robust to the number of irrelevant prompts used for decentering. We further note that even without the decentering step, MEMOIR still demonstrates state-of-the-art performance with a large performance margin to previous methods.

Table 11: Effect of varying the number of irrelevant prompts for activation centering on three post-edit metrics. Results are for experiments with 1,000 edits on ZsRE dataset using LLaMA-3 model.

| Number of Irrelevant Prompts | Rel. | Gen. | Loc. | Avg. |
|---|---|---|---|---|
| 0 (no centering) | 0.92 | 0.85 | 1.00 | 0.92 |
| 10 | 0.93 | 0.85 | 1.00 | 0.93 |
| 100 | 0.94 | 0.85 | 1.00 | 0.93 |
| 1,000 | 0.93 | 0.85 | 1.00 | 0.93 |

## B.3 Additional ablations

**Impact of active index selection strategies** We adopt a TopHash strategy to select memory columns for updates. It consists of two key components: (1) selecting informative indices from the input activation to serve as fingerprints that help group semantically similar inputs, and (2) applying a hashing step that spreads the updates across diverse memory columns to prevent forgetting.

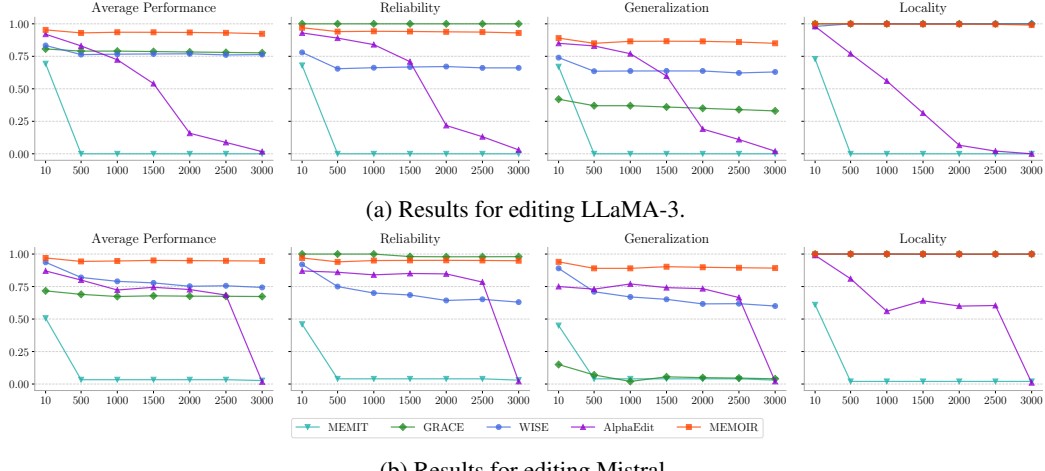

(a) Results for editing LLaMA-3.

(b) Results for editing Mistral.

Figure 7: Performance v.s. different number of edits on ZsRE: (a) Results for LLaMA-3. (b) Results for Mistral. The x-axis denotes the number of sequential edits applied, and the y-axis shows the corresponding metric value.

Using only *TopK* indices for those with the highest activation magnitudes tends to concentrate updates on a small subset of important parameters. This behavior risks degrading performance by repeatedly overwriting critical knowledge of previous edits. On the other hand, selecting indices purely at *Random* significantly reduces performance: such selections lack semantic grounding and often fail to activate relevant information. For example, even when a previously edited training sample is encountered, random selection frequently targets irrelevant parameters, resulting in poor reliability.

A more structured alternative is sample-specific *Hash*, where a fixed random mask is deterministically assigned to each input. This method ensures that repeated inputs activate the same memory locations, increasing reliability for seen examples. However, this strategy fails to generalize to paraphrased inputs. Since the hashing is based on exact input patterns rather than semantic content, it produces masks that are specific but semantically meaningless.

Table 12: Performance of different grouping strategies on three post-edit metrics. We rank each method according to their randomness level.

| Method | Selection Strategy | Rel. | Gen. | Loc. | Avg. |
|---|---|---|---|---|---|
| TopK | Largest | 0.87 | 0.82 | 1.00 | 0.90 |
| Random | Random | 0.31 | 0.30 | 1.00 | 0.54 |
| Hash | Hash | 0.95 | 0.31 | 1.00 | 0.75 |
| **MEMOIR (TopHash)** | Largest | 0.94 | 0.85 | 1.00 | **0.93** |

To address this, we apply a fixed permutation to hash the selected indices. TopHash exploits semantic similarity in the activations by selecting informative indices, and then mapping them to different memory columns based on a fixed permutation.

Table 12 compares TopHash with the previously mentioned alternatives, namely *TopK*, *Random* and *Hash* methods. As shown in the table, both TopK and Random baselines underperform. TopK alone harms performance by repeatedly updating key parameters. Random selection fails in both reliability and generalization, as it always activates unrelated memory locations. Sample-specific hashing can memorize training samples but lacks semantic sensitivity, limiting generalization. The success of TopHash highlights the importance of (1) selecting distinctive, semantically meaningful indices, and (2) distributing updates through a consistent hashing mechanism.

## B.4 Editing Scalability with Increasing Number of Edits

We present in Figure 7 the editing performance of MEMOIR as the number of total edits increases from 10 to 3000, in comparison with existing baselines. The results are obtained by applying edits to both LLaMA-3 and Mistral models on samples from the ZsRE dataset, providing a more detailed view complementing the trends shown in Figure 1. As shown in Figure 7, MEMOIR consistently outperforms all competing approaches in terms of average performance across reliability, generalization, and locality.

We observe that parametric editing methods such as AlphaEdit and MEMIT exhibit a clear degradation in performance as the number of edits increases, indicating susceptibility to forgetting. In contrast, `MEMOIR` maintains stable performance across all edit scales. Although GRACE achieves slightly better reliability scores, this comes at the cost of generalization, due to its rigid memorization mechanism that fails to adapt semantically. Overall, when averaged across all metrics, `MEMOIR` offers the most balanced and robust performance, demonstrating strong scalability and resilience under sequential editing.

## B.5 Evaluation on RippleEdits

Table 13: Example from the POPULAR dataset (multi-hop reasoning task).

| POPULAR dataset | |
|---|---|
| Prompt $x_e$ | The name of the country which Academy Award for Best Picture is associated with is |
| Original target $y$ | United States of America |
| Edit target $y_e$ | Wassoulou Empire |
| Prompt (Logic Generalization) $x_{gen}$ | The name of the continent which Academy Award for Best Picture is part of is |
| Target (Logic Generalization) $y_{gen}$ | Africa |
| Prompt (Relation Specificity) $x_{spec}$ | The name of the award Academy Award for Best Picture won is |
| Target (Relation Specificity) $y_{spec}$ | National Board of Review Award for Best Film |

We conduct experiments on the POPULAR dataset (illustrated in Table 13) in the RippleEdits benchmark [37] under sequential-edit settings with $T = 10$ and $T = 100$ using LLaMA-3, using three metrics: (i) Reliability (Rel.), the model's ability to absorb newly injected knowledge; (ii) Logic Generalization (Gen.), which evaluates whether the edit supports consistent logical inference over related facts; and (iii) Relation Specificity (Spec.), measuring whether unrelated relations remain unaffected. To ensure comparability, we filter the dataset to obtain 10 and 100 edit subsets containing all Rel., Gen., and Spec. cases. These metrics reflect more complex forms of generalization and locality compared to standard factual accuracy. All evaluations are performed in a lifelong editing setup and measured after the full sequence of edits.

We present the multi-hop reasoning results on RippleEdits dataset in Table 14. Despite the difficulty of this regime, as evidenced by consistently low Gen. and Spec. scores across all baselines, `MEMOIR` attains the highest performance on 5 out of 6 metrics and achieves the best average accuracy across all three tasks. These results demonstrate its strong capacity for generalization even under challenging sequential-edit settings. These findings position `MEMOIR` as a solid foundation for advancing multi-hop generalization.

Table 14: Performance on RippleEdits (POPULAR dataset). MEMOIR achieves consistently strong reliability while also improving logical generalization and specificity relative to prior methods.

| | $T = 10$ | | | | $T = 100$ | | | |
|---|---|---|---|---|---|---|---|---|
| Method | Rel. | Gen. | Spec. | Avg. | Rel. | Gen. | Spec. | Avg. |
| ROME [11] | 0.80 | 0.04 | 0.29 | 0.38 | 0.01 | 0.01 | 0.02 | 0.01 |
| MEMIT [14] | 0.98 | 0.27 | 0.29 | 0.51 | 0.01 | 0.00 | 0.01 | 0.01 |
| GRACE [8] | 1.00 | 0.00 | 0.05 | 0.35 | 0.96 | 0.01 | 0.10 | 0.36 |
| AlphaEdit [15] | 1.00 | 0.13 | 0.44 | 0.52 | 0.90 | 0.22 | 0.46 | 0.53 |
| WISE [9] | 0.78 | 0.02 | 0.04 | 0.28 | 0.68 | 0.10 | 0.06 | 0.28 |
| **MEMOIR (Ours)** | 1.00 | 0.27 | 0.54 | **0.60** | 0.98 | 0.20 | 0.59 | **0.59** |

