# OpenReview forum: "MEMOIR: Lifelong Model Editing with Minimal Overwrite and Informed Retention for LLMs"
_NeurIPS.cc/2025/Conference — NeurIPS 2025 poster_

### Official Review · Reviewer_DSuA · 2025-06-14

**Clarity:** 3
**Significance:** 3
**Originality:** 3
**Rating:** 5
**Confidence:** 4

**Summary:**

This paper introduces MEMOIR (Model Editing with Minimal Overwrite and Information Retention), a lifelong model editing method that affords a good trade-off between reliability, generalization, and locality. The edits are performed through a separate fully-connected
layer whose output is later added to that of the original model layer. These edits are restricted to subsets of parameter selected by an input-dependent sparse mask. This allows MEMOIR to mitigate catastrophic forgetting. At inference time, they use the sparse mask to find the closest edited prompt to the input. The model's predictions on non-edited prompts are preserved. They evaluate with 4 models and compare with a wide range of baselines.

**Questions:**

1 - How does MEMOIR compare to prior methods on benchmarks (for example https://arxiv.org/abs/2307.12976) that measures the impact of fact updates on other facts?

2 - Do you expect your approach to generalize to bigger models?

**Ethical Concerns:**

["NO or VERY MINOR ethics concerns only"]

**Final Justification:**

The authors have answered all the questions and addressed what I considered the main weakness of the paper. This additional experiment should be included in the final version of the paper.

**Limitations:**

yes

**Quality:**

4

**Strengths And Weaknesses:**

Strengths:
- The method offers a good balance between the main metrics of interest—reliability, generalization, and locality—without introducing additional severe computational overload. The memory module is non-invasive and does not lead to catastrophic forgetting.
- MEMOIR is compared to an extensive baseline.
- The paper is well-structured, easy to read.

Weaknesses:
- An important evaluation of model editing methods was not mentioned in the paper. Updating some facts can affect other facts in the model that are not necessarily captured by semantic similarity. There are already existing benchmarks such as this one https://arxiv.org/abs/2307.12976 to analyze this phenomenon.

Minor comments:
There are 2 'the' on line 316.

---

> ### Author Rebuttal · Authors · 2025-07-31
>
> **General comment**
>
> First, we would like to respectfully bring to the reviewer’s attention a correction in our reported results due to a technical oversight. Specifically, when computing TopHash indices during inference, we inadvertently included label information by using features of the concatenated [prompt, label] sequence.
>
> We have corrected this by recomputing TopHash indices using centered features averaged over prompt tokens only. This affected the semantic representation used for similarity computations and the subsequent routing step, impacting mainly the generalization metric for ZsRE QA task with LLaMA 3. In contrast, reliability and locality are only minimally affected. We see no or negligible degradation in the remaining combinations of benchmarks and models across all metrics.
>
> The revised results are provided below. Importantly, none of our previous conclusions are affected. In particular, in all settings, MEMOIR performance **remains state-of-the-art**, consistently outperforming all baselines across all metrics with a strong margin over the second-best methods.
>
> **Table 1: Q&A task results.**
> |Method|T=1||||T=10||||T=100||||T=1000||||
> |-|-|-|-|-|-|-|-|-|-|-|-|-|-|-|-|-|
> ||Rel.|Gen.|Loc.|Avg.|Rel.|Gen.|Loc.|Avg.|Rel.|Gen.|Loc.|Avg.|Rel.|Gen.|Loc.|Avg.|
> |LLaMA-3-8B|
> |MEMOIR(before)|1.00|0.98|1.00|0.99|0.99|0.96|1.00|0.98|0.97|0.94|1.00|0.97|0.95|0.91|1.00|0.95|
> |GRACE|1.00|0.46|1.00|0.82|1.00|0.42|1.00|0.81|1.00|0.39|1.00|0.80|1.00|0.37|1.00|0.79|
> |WISE|0.92|0.84|1.00|0.92|0.78|0.74|0.98|0.83|0.62|0.60|1.00|0.74|0.66|0.64|1.00|0.77|
> |AlphaEdit|0.98|0.89|1.00|0.96|0.93|0.85|0.98|0.92|0.91|0.79|0.94|0.88|0.84|0.77|0.56|0.72|
> |**MEMOIR**|1.00|1.00|1.00|**1.00**|0.97|0.89|1.00|**0.95**|0.96|0.89|1.00|**0.95**|0.94|0.85|1.00|**0.93**|
> |Mistral-7B|
> |MEMOIR(before)|1.00|0.93|1.00|0.98|0.98|0.91|1.00|0.96|0.96|0.89|1.00|0.95|0.93|0.87|1.00|0.93|
> |GRACE|1.00|0.36|1.00|0.79|1.00|0.15|1.00|0.72|1.00|0.15|1.00|0.72|1.00|0.02|1.00|0.67|
> |WISE|0.98|0.97|1.00|0.98|0.92|0.89|1.00|0.94|0.87|0.80|1.00|0.89|0.70|0.67|1.00|0.79|
> |AlphaEdit|0.83|0.77|1.00|0.87|0.87|0.75|0.99|0.87|0.86|0.74|0.95|0.85|0.85|0.72|0.68|0.75|
> |**MEMOIR**|1.00|0.99|1.00|**1.00**|0.97|0.94|1.00|**0.97**|0.95|0.91|1.00|**0.95**|0.94|0.89|1.00|**0.94**|
>
> **Table 2: Hallucination correction task results.**
> ||LLaMA-3-8B |||||||| Mistral-7B ||||||||
> |-|-|-|-|-|-|-|-|-|-|-|-|-|-|-|-|-|
> ||T=1||T=10||T=100||T=600||T=1||T=10||T=100||T=600||
> |Method|Rel.|Loc.|Rel.|Loc.|Rel.|Loc.|Rel.|Loc.|Rel.|Loc.|Rel.|Loc.|Rel.|Loc.|Rel.|Loc.|
> |MEMOIR(before)|1.00|1.00|1.01|1.00|1.09|1.00|1.37|1.00|1.00|1.00|1.02|1.00|1.09|1.00|1.22|1.00|
> |GRACE|1.05|1.00|7.10e1|1.00|7.12e1|1.00|7.73e1|1.00|1.39|1.00|5.97|1.00|9.53|1.00|9.57|1.00|
> |WISE|4.93e1|0.98|1.46|0.95|2.10|0.99|3.20|0.99|1.40|1.00|2.56|0.94|1.31|0.99|5.21|0.93|
> |AlphaEdit|1.58|1.00|3.12|0.98|5.97|0.93|8.49e3|0.05|1.75|1.00|1.76|1.00|2.87|0.98|1.70e2|0.88|
> |**MEMOIR**|1.00|1.00|1.01|1.00|1.07|1.00|1.25|1.00|1.00|1.00|1.02|1.00|1.09|1.00|1.22|1.00|
>
> **Rebuttal**
>
> We thank the reviewer DSuA for their valuable feedback. We are pleased they find our paper well-structured and easy to read, our baseline comparison extensive, and our method offering a good balance between the main metrics.  Below, we address their specific comments and questions.
>
> **W1/Q1: Evaluation on RippleEdits**
> We thank the reviewer for drawing our attention to this benchmark. First, we note that, in the current manuscript, MEMOIR follows the evaluation protocols established in prior work (ROME, MEMIT, GRACE, WISE) and covers a broad range of factual knowledge editing tasks, including question answering (ZsRE), hallucination correction (SelfCheckGPT), and OOD generalization (temporal dataset).
>
> We recognize that evaluating how edited facts influence other facts beyond semantic similarity is a compelling direction that would further strengthen our contribution. Following the reviewer’s suggestion, we have added **new experiments** on the RippleEdits benchmark to analyze how MEMOIR impacts surrounding factual knowledge during the editing process.
>
> We evaluate MEMOIR on the POPULAR dataset under 10 and 100 sequential-edit settings, using three metrics: Reliability (Rel.), Logic Generalization (Gen.), and Relation Specificity (Spec.). Rel. quantifies the model’s ability to absorb new knowledge (as defined in the draft), Gen. captures whether the edit supports consistent logical inference over related facts, and Spec. verifies that unrelated relations remain unchanged. These metrics reflect more complex forms of generalization and locality. The results are presented in the table below. Notice that evaluations are based on our current implementation under a lifelong editing setup. In particular, they are performed only after all edits in the sequence.
>
> |Method|T=10||||T=100||||
> |-|-|-|-|-|-|-|-|-|
> ||Rel.|Gen.|Spec.|Avg.|Rel.|Gen.|Spec.|Avg.|
> |ROME|0.80|0.04|0.29|0.38|0.01|0.01|0.02|0.01|
> |MEMIT|0.98|0.27|0.29|0.51|0.01|0.00|0.01|0.01|
> |GRACE|1.00|0.00|0.05|0.35|0.96|0.01|0.10|0.36|
> |AlphaEdit|1.00|0.13|0.44|0.52|0.90|0.22|0.46|0.53|
> |WISE|0.78|0.02|0.04|0.28|0.68|0.10|0.06|0.28|
> |**MEMOIR**|1.00|0.27|0.54|**0.60**|0.98|0.20|0.59|**0.59**|
>
> Despite this being a highly challenging regime, as reflected by lower Gen. and Spec. scores across all baselines,  MEMOIR achieves the highest performance on 5 out of 6 metrics and obtains the best average accuracy across all 3 tasks. This demonstrates its strong generalization ability even in this challenging setting. While there remains room for overall improvement, these results highlight MEMOIR as a strong foundation for advancing multi-hop generalization. Additionally, we expect further refinements, including improved training and tuning over hyperparameters, to yield stronger results on this benchmark in the revised version of our work.
> We thank the reviewer for directing us to this new benchmark, and we will update the manuscript to include these new results to further strengthen our contribution, along with a dedicated paragraph analyzing and discussing them in depth. We kindly ask the reviewer to reconsider their score based on the new evaluation results.
>
> **Q2: Generalizability to larger models**
> We follow prior works in the knowledge editing field and have covered major LLMs experimented in this field with varying sizes and architectures, including GPT-j-6B, LLaMA-2-7B, Mistral-7B, LLaMA-3-8B.
> We expect our method to generalize well to even larger models for the following reasons:
>
> 1. The residual memory, initialized from an intermediate FFN projection layer, contains more parameters in larger models. Assuming a fixed active subspace per edit, this enables storing more edits with reduced interference and less forgetting during sequential editing.
> 2. Larger models capture semantic variation more robustly and are expected to generalize more effectively across paraphrases. Our conditional knowledge activation mechanism can thus more accurately identify semantically relevant and irrelevant samples during inference.
>
> We hope this effectively addresses the reviewer’s concerns and sincerely appreciate their positive assessment. We remain available for any further discussion or clarification.

---

> > ### Comment · Reviewer_DSuA · 2025-08-03
> >
> > Thank you for your answers and further clarifications. The authors have answered all the questions and addressed the main weakness. I will update the score accordingly. Even though previous work did not evaluate on RippleEdits to investigate the impact of fact updates on other facts, this experiment should be included in the paper as it might be very informative for the community.

---

### Official Review · Reviewer_HSS5 · 2025-07-02

**Clarity:** 3
**Significance:** 2
**Originality:** 3
**Rating:** 4
**Confidence:** 4

**Summary:**

This paper proposes MEMOIR, a framework for lifelong model editing in large language models (LLMs), designed to inject new knowledge while minimizing interference with existing knowledge. MEMOIR introduces a residual memory module with sparse, data-dependent masking, allowing each edit to target a distinct subset of parameters. During inference, MEMOIR activates memory selectively using a similarity-based routing mechanism, ensuring generalization to rephrased queries and suppression of irrelevant memory usage. The method is evaluated across tasks such as question answering, hallucination correction, and OOD generalization, showing good trade-offs between reliability, generalization, and locality, scaling effectively to thousands of sequential edits. The framework outperforms previous parametric and non-parametric methods including MEMIT, AlphaEdit, and WISE.

**Questions:**

Have you experimented with extending MEMOIR to multiple transformer layers, or hierarchically allocating memory? Would it improve generalization or allow for more complex edits?

How sensitive is performance to the choice of routing threshold? Could it be learned during editing, rather than heuristically set?

In practice, if the number of edits grows indefinitely, the residual memory may become large. Have you considered any memory pruning or compression mechanisms?

**Ethical Concerns:**

["NO or VERY MINOR ethics concerns only"]

**Final Justification:**

The authors answered all my questions and addressed my concerns. I updated my score to 4 and recommend an acceptance.

**Limitations:**

The authors appropriately note limitations in scope (single-layer editing, decoder-only models) and acknowledge future directions. The societal impact of memory-based editing (e.g., misinformation correction vs. malicious manipulation) is not deeply discussed but could be strengthened with a short reflection.

**Quality:**

3

**Strengths And Weaknesses:**

**Strength**

The paper is methodologically solid. The empirical evaluation is extensive, covering multiple models, tasks, and a wide range of edit scales. The performance gains are clearly demonstrated with ablation studies.

The paper is clearly written, with intuitive figures that help illustrate the core mechanisms. The explanation of the masking strategy (TopHash) and inference-time routing is particularly clear and technically precise.

The combination of sparse memory allocation and conditional activation based on TopHash is novel in the context of LLM editing. While sparsity and activation gating are known techniques, their integration into this framework and use for routing edits in a memory-efficient and interference-minimizing way is well-motivated.

**Weakness**

MEMOIR only modifies a single transformer layer’s projection weights. While this suffices for many factual edits, it may limit applicability to complex or multi-hop knowledge edits requiring broader changes across multiple layers.

The effectiveness of TopHash-based routing depends on the number of active indices and the similarity threshold. Though the paper provides ablations, further justification or principled selection strategies would improve robustness and usability.

---

> ### Author Rebuttal · Authors · 2025-07-31
>
> **General comment**
>
> First, we would like to respectfully bring to the reviewer’s attention a correction in our reported results due to a technical oversight. Specifically, when computing TopHash indices during inference, we inadvertently included label information by using features of the concatenated [prompt, label] sequence.
>
> We have corrected this by recomputing TopHash indices using centered features averaged over prompt tokens only. This affected the semantic representation used for similarity computations and the subsequent routing step, impacting mainly the generalization metric for ZsRE QA task with LLaMA 3. In contrast, reliability and locality are only minimally affected. We see no or negligible degradation in the remaining combinations of benchmarks and models across all metrics.
>
> The revised results are provided below. Importantly, none of our previous conclusions are affected. In particular, in all settings, MEMOIR performance **remains state-of-the-art**, consistently outperforming all baselines across all metrics with a strong margin over the second-best methods.
>
> **Table 1: Q&A task results.**
> |Method|T=1||||T=10||||T=100||||T=1000||||
> |-|-|-|-|-|-|-|-|-|-|-|-|-|-|-|-|-|
> ||Rel.|Gen.|Loc.|Avg.|Rel.|Gen.|Loc.|Avg.|Rel.|Gen.|Loc.|Avg.|Rel.|Gen.|Loc.|Avg.|
> |LLaMA-3-8B|
> |MEMOIR(before)|1.00|0.98|1.00|0.99|0.99|0.96|1.00|0.98|0.97|0.94|1.00|0.97|0.95|0.91|1.00|0.95|
> |GRACE|1.00|0.46|1.00|0.82|1.00|0.42|1.00|0.81|1.00|0.39|1.00|0.80|1.00|0.37|1.00|0.79|
> |WISE|0.92|0.84|1.00|0.92|0.78|0.74|0.98|0.83|0.62|0.60|1.00|0.74|0.66|0.64|1.00|0.77|
> |AlphaEdit|0.98|0.89|1.00|0.96|0.93|0.85|0.98|0.92|0.91|0.79|0.94|0.88|0.84|0.77|0.56|0.72|
> |**MEMOIR**|1.00|1.00|1.00|**1.00**|0.97|0.89|1.00|**0.95**|0.96|0.89|1.00|**0.95**|0.94|0.85|1.00|**0.93**|
> |Mistral-7B|
> |MEMOIR(before)|1.00|0.93|1.00|0.98|0.98|0.91|1.00|0.96|0.96|0.89|1.00|0.95|0.93|0.87|1.00|0.93|
> |GRACE|1.00|0.36|1.00|0.79|1.00|0.15|1.00|0.72|1.00|0.15|1.00|0.72|1.00|0.02|1.00|0.67|
> |WISE|0.98|0.97|1.00|0.98|0.92|0.89|1.00|0.94|0.87|0.80|1.00|0.89|0.70|0.67|1.00|0.79|
> |AlphaEdit|0.83|0.77|1.00|0.87|0.87|0.75|0.99|0.87|0.86|0.74|0.95|0.85|0.85|0.72|0.68|0.75|
> |**MEMOIR**|1.00|0.99|1.00|**1.00**|0.97|0.94|1.00|**0.97**|0.95|0.91|1.00|**0.95**|0.94|0.89|1.00|**0.94**|
>
> **Table 2: Hallucination correction task results.**
> ||LLaMA-3-8B |||||||| Mistral-7B ||||||||
> |-|-|-|-|-|-|-|-|-|-|-|-|-|-|-|-|-|
> ||T=1||T=10||T=100||T=600||T=1||T=10||T=100||T=600||
> |Method|Rel.|Loc.|Rel.|Loc.|Rel.|Loc.|Rel.|Loc.|Rel.|Loc.|Rel.|Loc.|Rel.|Loc.|Rel.|Loc.|
> |MEMOIR(before)|1.00|1.00|1.01|1.00|1.09|1.00|1.37|1.00|1.00|1.00|1.02|1.00|1.09|1.00|1.22|1.00|
> |GRACE|1.05|1.00|7.10e1|1.00|7.12e1|1.00|7.73e1|1.00|1.39|1.00|5.97|1.00|9.53|1.00|9.57|1.00|
> |WISE|4.93e1|0.98|1.46|0.95|2.10|0.99|3.20|0.99|1.40|1.00|2.56|0.94|1.31|0.99|5.21|0.93|
> |AlphaEdit|1.58|1.00|3.12|0.98|5.97|0.93|8.49e3|0.05|1.75|1.00|1.76|1.00|2.87|0.98|1.70e2|0.88|
> |**MEMOIR**|1.00|1.00|1.01|1.00|1.07|1.00|1.25|1.00|1.00|1.00|1.02|1.00|1.09|1.00|1.22|1.00|
>
> **Rebuttal**
>
>
> We thank Reviewer HSS5 for their valuable feedback. We are pleased that they found our paper well-motivated, clearly written, our method both solid and novel, and the empirical evaluation extensive. Below, we address their specific comments and questions.
>
> **W1/Q1: Editing a single FFN projection layer**
> Although arguably the simplest design choice, in our experiments, we note that editing a single FFN projection layer is shown to be highly effective for MEMOIR. Crucially, it achieves significantly higher performance than prior methods that edit multiple layers (MEMIT, AlphaEdit), while using 5x fewer trainable parameters, resulting in much lower computational cost. We believe this is a major strength of our method since it combines improved performance with significantly fewer trainable parameters.
> The reviewer’s suggestion to extend to multiple layers is indeed a promising direction, which we leave for future work. Indeed, it’s reasonable to expect such an extension to enhance generalization in tasks involving multi-hop reasoning and allow for more complex edits.
>
> Furthermore, we kindly refer the reviewer to our response to reviewer DSuA, where we present **a new experiment** evaluating MEMOIR on RippleEffects[1], a benchmark designed to assess two-hop reasoning in knowledge editing. Notably, despite editing only a single layer, MEMOIR achieves the strongest generalization performance in this multi-hop reasoning setting, outperforming baseline methods that modify multiple layers, such as MEMIT and AlphaEdit.
>
> **W2/Q2: Robustness of TopHash-based routing**
> We selected the hyperparameters for the number of active indices $k$ and mask similarity threshold based on performance over 1000 edits on ZsRE. As shown below, both parameters $\tau$ and $k$ remain highly robust across varying edit counts, and generalize well to other tasks.
> In MEMOIR, $k$ controls the trade-off between edit specificity and interference: larger $k$better captures individual edits, while smaller $k$ reduces forgetting. Despite this trade-off, MEMOIR remains stable across a wide range of $k$: we use the same $k = 4096$ across different model architectures (LLaMA-2, LLaMA-3, Mistral), tasks (ZsRE QA, SelfCheckGPT), and edit scales (from 1 to 7000 edits). As shown in Figure 6, the reliability remains high for $k$ between $[1024, 8192]$, indicating strong robustness and ease of tuning. Similarly, $\tau$ is set once per model to reflect differences in representation spaces and remains fixed across all tasks and edit counts. This consistent performance demonstrates the generalizability of our TopHash-based routing strategy.
>
> **Q2: Learning a mask similarity threshold**
> We thank the reviewer for the suggestion to introduce a learnable thresholding mechanism. We did not adopt it because we want to keep the simplicity of our method with minimal requirements. Indeed, MEMOIR has demonstrated substantial performance gains over all baselines, including the ones that introduce a learnable router mechanism (e.g., WISE and MEND). However, the use of an adaptive router assumes the availability of semantically aligned or paraphrased prompts during training, which introduces extra computational costs and limits the adaptability of the method. However, learnable routing is a promising addition that could further strengthen MEMOIR’s strong generalization across irrelevant prompts, which we leave for future works.
>
> **Q3: Scalability to growing number of edits**
> In terms of the **scalability of the memory footprint**, the residual memory, initialized as a clone of the original FFN projection layer, has a fixed size and does not grow with the number of edits. Precisely, MEMOIR only requires a copy of a single linear layer in a LLM and one binary (rather than float32) mask per edit. Specifically, for a LLaMA-3-8B model and 7000 edits, this means a memory increase of 0.7% and 0.1% for parameters and masks, respectively, resulting in an overall increase of 0.8%.
>
> In terms of the **scalability of performance**, a key feature of MEMOIR is its ability to incorporate a large number of edits with minimal forgetting. This is achieved through the TopHash knowledge distribution mechanism, which mitigates catastrophic forgetting during sequential edits. Empirically, MEMOIR maintains strong performance even with 7000 edits, showing no obvious degradation. To further illustrate the scalability of MEMOIR to a large number of edits, we conducted additional evaluations of MEMOIR with 11k and 15k edits on ZsRE QA dataset. As shown in the table, MEMOIR maintains stable performance with no collapse up to 15k edits. These experiments showcase the superior robustness of MEMOIR to longer edits; while baselines suffer in one or even all metrics, MEMOIR maintains strong performance even for 15k edits.
>
> |Method|Rel.|Gen.|Loc.|Avg.|Rel.|Gen.|Loc.|Avg.|
> |-|-|-|-|-|-|-|-|-|
> ||11k||||15k||||
> |GRACE|0.99|0.28|1.00|0.76|0.99|0.28|1.00|0.76|
> |WISE|0.44|0.42|1.00|0.62|0.43|0.41|1.00|0.61|
> |AlphaEdit|0.01|0.01|0.00|0.01|0.02|0.02|0.00|0.01|
> |**MEMOIR**|0.89|0.81|0.99|**0.90**|0.87|0.78|0.98|**0.88**|
>
> Clearly, if the number of edits grows indefinitely, at a certain point it becomes infeasible to store all knowledge within a single memory. A possible improvement for such cases, inspired by the reviewer’s suggestion, is to store edits across multiple memories and compress them into a single memory before inference. This can be achieved by pruning redundancies within each memory and then merging them while minimizing interference, similar in spirit to the strategy proposed in Ties-Merging [2]. We believe this line of work is promising for scaling knowledge editing beyond current memory limits.
>
> **L1: Societal impact of memory-based editing**
> We appreciate the reviewer’s thoughtful observation on the societal implications of our method. Indeed, memory-based knowledge editing has far-reaching implications. On the positive side, it enables rapid correction of misinformation, dynamic adaptation to evolving world knowledge, and alignment with ethical standards, especially important for applications in education, healthcare, and safety-critical domains. However, as the reviewer rightly notes, these capabilities also raise concerns. The same techniques could be exploited for malicious purposes, such as inserting biased content, manipulating facts, or selectively erasing information to influence user beliefs or public opinion. Following the reviewer’s suggestion, we will include a dedicated section in the revised manuscript.
>
> We thank the reviewer again for their constructive feedback and hope our clarifications help address their concerns. We would appreciate a reconsideration of the score in light of these responses.
>
> [1] Evaluating the Ripple Effects of Knowledge Editing in Language Models, Cohen et al., TACL 2024.
>
> [2] TIES-Merging: Resolving Interference When Merging Models, Yadav et al., NeurIPS 2023.

---

### Official Review · Reviewer_ftpB · 2025-07-05

**Clarity:** 4
**Significance:** 3
**Originality:** 3
**Rating:** 5
**Confidence:** 5

**Summary:**

The paper introduces MEMOIR, a scalable framework for lifelong model editing that updates large language models (LLMs) with new knowledge while minimizing forgetting of previous edits. MEMOIR uses a residual memory module with sparse, data-dependent activations to isolate edits, combined with a conditional inference mechanism that activates only relevant memory for semantically similar prompts. Through experiments on tasks like question answering, hallucination correction, and out-of-distribution generalization using models such as LLaMA-3 and Mistral, MEMOIR shows superior performance in reliability, generalization, and locality, scaling up to 7000 edits with minimal performance degradation.

**Questions:**

This paper choose the threshold 𝜏 for determining whether a prompt is relevant to stored edits is fixed.
It might worthwhile to consider Introducing a learnable or context-sensitive thresholding mechanism (e.g., trained via meta-learning or reinforcement learning) to better distinguish semantically similar prompts from unrelated ones.

**Ethical Concerns:**

["NO or VERY MINOR ethics concerns only"]

**Final Justification:**

The author's rebuttal addressed all the three issues raised in my review, thus I have no further questions. I maintain the same score "Accept".

**Limitations:**

yes

**Quality:**

4

**Strengths And Weaknesses:**

Strengths:

MEMOIR addresses a fundamental challenge in LLMs—efficient post-hoc knowledge editing—with a well-motivated, technically sound approach. By sparsifying activation via TopHash and storing updates in a dedicated residual memory, the framework avoids catastrophic forgetting while supporting generalization to paraphrased queries. Its conditional knowledge activation at inference is novel, precise, and efficient, outperforming strong baselines (e.g., WISE, AlphaEdit) across multiple metrics and LLM architectures. The scalability to 7000 sequential edits with stable performance demonstrates practical robustness.

The paper is thorough in empirical evaluation, comparing MEMOIR to a broad set of baselines across diverse tasks. It supports claims with detailed ablations (e.g., removing conditional activation or varying sparsity levels), showing clear advantages of the proposed design. The clarity of writing, theoretical intuitions, and visualizations (e.g., mask overlap distributions and reliability curves) further enhance the paper’s readability and impact.

Weaknesses:
Despite its strengths, MEMOIR is still limited by its scope. It edits only a single FFN projection layer, which may restrict its ability to handle deeply integrated or abstract knowledge. This could be problematic for tasks involving multi-hop reasoning or systemic model behaviors. Additionally, the TopHash-based sparsity strategy, while effective, introduces a fixed permutation mechanism that may lack adaptiveness, and its dependence on mask similarity thresholds might not generalize robustly across all tasks or model types.

---

> ### Author Rebuttal · Authors · 2025-07-31
>
> **General comment**
>
> First, we would like to respectfully bring to the reviewer’s attention a correction in our reported results due to a technical oversight. Specifically, when computing TopHash indices during inference, we inadvertently included label information by using features of the concatenated [prompt, label] sequence.
>
> We have corrected this by recomputing TopHash indices using centered features averaged over prompt tokens only. This affected the semantic representation used for similarity computations and the subsequent routing step, impacting mainly the generalization metric for ZsRE QA task with LLaMA 3. In contrast, reliability and locality are only minimally affected. We see no or negligible degradation in the remaining combinations of benchmarks and models across all metrics.
>
> The revised results are provided below. Importantly, none of our previous conclusions are affected. In particular, in all settings, MEMOIR performance **remains state-of-the-art**, consistently outperforming all baselines across all metrics with a strong margin over the second-best methods.
>
> **Table 1: Q&A task results.**
> |Method|T=1||||T=10||||T=100||||T=1000||||
> |-|-|-|-|-|-|-|-|-|-|-|-|-|-|-|-|-|
> ||Rel.|Gen.|Loc.|Avg.|Rel.|Gen.|Loc.|Avg.|Rel.|Gen.|Loc.|Avg.|Rel.|Gen.|Loc.|Avg.|
> |LLaMA-3-8B|
> |MEMOIR(before)|1.00|0.98|1.00|0.99|0.99|0.96|1.00|0.98|0.97|0.94|1.00|0.97|0.95|0.91|1.00|0.95|
> |GRACE|1.00|0.46|1.00|0.82|1.00|0.42|1.00|0.81|1.00|0.39|1.00|0.80|1.00|0.37|1.00|0.79|
> |WISE|0.92|0.84|1.00|0.92|0.78|0.74|0.98|0.83|0.62|0.60|1.00|0.74|0.66|0.64|1.00|0.77|
> |AlphaEdit|0.98|0.89|1.00|0.96|0.93|0.85|0.98|0.92|0.91|0.79|0.94|0.88|0.84|0.77|0.56|0.72|
> |**MEMOIR**|1.00|1.00|1.00|**1.00**|0.97|0.89|1.00|**0.95**|0.96|0.89|1.00|**0.95**|0.94|0.85|1.00|**0.93**|
> |Mistral-7B|
> |MEMOIR(before)|1.00|0.93|1.00|0.98|0.98|0.91|1.00|0.96|0.96|0.89|1.00|0.95|0.93|0.87|1.00|0.93|
> |GRACE|1.00|0.36|1.00|0.79|1.00|0.15|1.00|0.72|1.00|0.15|1.00|0.72|1.00|0.02|1.00|0.67|
> |WISE|0.98|0.97|1.00|0.98|0.92|0.89|1.00|0.94|0.87|0.80|1.00|0.89|0.70|0.67|1.00|0.79|
> |AlphaEdit|0.83|0.77|1.00|0.87|0.87|0.75|0.99|0.87|0.86|0.74|0.95|0.85|0.85|0.72|0.68|0.75|
> |**MEMOIR**|1.00|0.99|1.00|**1.00**|0.97|0.94|1.00|**0.97**|0.95|0.91|1.00|**0.95**|0.94|0.89|1.00|**0.94**|
>
> **Table 2: Hallucination correction task results.**
> ||LLaMA-3-8B |||||||| Mistral-7B ||||||||
> |-|-|-|-|-|-|-|-|-|-|-|-|-|-|-|-|-|
> ||T=1||T=10||T=100||T=600||T=1||T=10||T=100||T=600||
> |Method|Rel.|Loc.|Rel.|Loc.|Rel.|Loc.|Rel.|Loc.|Rel.|Loc.|Rel.|Loc.|Rel.|Loc.|Rel.|Loc.|
> |MEMOIR(before)|1.00|1.00|1.01|1.00|1.09|1.00|1.37|1.00|1.00|1.00|1.02|1.00|1.09|1.00|1.22|1.00|
> |GRACE|1.05|1.00|7.10e1|1.00|7.12e1|1.00|7.73e1|1.00|1.39|1.00|5.97|1.00|9.53|1.00|9.57|1.00|
> |WISE|4.93e1|0.98|1.46|0.95|2.10|0.99|3.20|0.99|1.40|1.00|2.56|0.94|1.31|0.99|5.21|0.93|
> |AlphaEdit|1.58|1.00|3.12|0.98|5.97|0.93|8.49e3|0.05|1.75|1.00|1.76|1.00|2.87|0.98|1.70e2|0.88|
> |**MEMOIR**|1.00|1.00|1.01|1.00|1.07|1.00|1.25|1.00|1.00|1.00|1.02|1.00|1.09|1.00|1.22|1.00|
>
>
> **Rebuttal**
>
> We thank reviewer ftpB for their valuable feedback. We appreciate that they find our paper well-motivated, technically sound, thorough in empirical evaluation, and with strong performance. Below, we address their specific comments and questions.
>
> **W1: Editing a single FFN projection layer**
> Although arguably the simplest design choice, our experiments show that editing a single FFN projection layer is highly effective for MEMOIR. Crucially, it achieves significantly higher performance than prior methods that edit multiple layers (MEMIT, AlphaEdit), while using 5x fewer trainable parameters, resulting in much lower computational cost. We believe this is a major strength of our method since it combines improved performance with significantly fewer trainable parameters.
> The reviewer’s suggestion to extend to multiple layers is indeed a promising direction, which we leave for future work. Indeed, it’s reasonable to expect such an extension to enhance performance in tasks involving multi-hop reasoning or systematic model behaviors.
>
> Furthermore, we kindly refer the reviewer to our response to reviewer DSuA, where we present **a new experiment** evaluating MEMOIR on RippleEffects[1], a benchmark designed to assess two-hop reasoning in knowledge editing. Notably, despite editing only a single layer, MEMOIR achieves the strongest generalization performance in this multi-hop reasoning setting, outperforming baseline methods that modify multiple layers, such as MEMIT and AlphaEdit.
>
> **Adaptability of the TopHash mechanism**
> In MEMOIR, TopHash distributes parameter updates across edits. The fixed permutation mechanism is tasked to redistribute the active indices from the influential parameters to less critical ones, which has been shown to be highly effective in reducing catastrophic forgetting. We note that this mechanism is highly adaptable across diverse settings. We apply TopHash across multiple model types (LLaMA-2, LLaMA-3, Mistral, GPT-J), tasks (ZsRE QA, SelfCheckGPT hallucination correction, temporal OOD generalization), and scales (from a single edit up to 7000 edits), where it consistently demonstrates strong performance and adaptability.
>
> **W1/Q1: Generalization of mask similarity threshold**
> The mask similarity threshold $\tau$ is used to determine whether a given prompt is semantically relevant to previously edited prompts. In our experiments, $\tau$ varies only across different model architectures due to intrinsic differences in their representation spaces. For each model, we apply a fixed $\tau$ across all experiments, including different tasks (ZsRE QA and SelfCheckGPT hallucination correction) and varying numbers of edits (from 1 to 7000), demonstrating its robustness and generalization across diverse testing scenarios.
>
> While we appreciate this suggestion to introduce a learnable thresholding mechanism, we opted for a simpler design to ensure minimal requirements and broad applicability.  Indeed, MEMOIR has demonstrated substantial performance gains over all baselines, including the ones that introduce a learnable router mechanism (e.g., WISE learns it with contrastive learning, and MEND learns it with meta-learning). However, the use of an adaptive router assumes the availability of semantically aligned or paraphrased prompts during training, which introduces extra computational costs and limits the adaptability of the method. Yet, learnable routing is a promising addition that could further strengthen MEMOIR’s strong generalization across irrelevant prompts, which we leave for future work.
>
> We hope this effectively addresses the reviewer’s concerns and sincerely appreciate their positive assessment. We remain available for any further discussion or clarification.
>
> [1] Evaluating the Ripple Effects of Knowledge Editing in Language Models, Cohen et al., TACL 2024.

---

### Official Review · Reviewer_8v8C · 2025-07-15

**Clarity:** 3
**Significance:** 3
**Originality:** 3
**Rating:** 4
**Confidence:** 3

**Summary:**

In this paper authors introduce MEMOIR in which they introduces a memory module, i.e., a dedicated fully-connected layer in a single transformer block where all edits are performed. MEMOIR mitigates catastrophic forgetting by allocating distinct parameter subsets to each edit and retrieving them during inference to activate only the relevant knowledge for a given prompt. They propose a structured sparsification of the input to this layer, dynamically activating only a sample-specific subset of parameters in the introduced memory module in the forward pass. Experiments on question answering, hallucination correction, and out-of-distribution generalization benchmarks across LLaMA-3 and Mistral demonstrate that MEMOIR achieves state-of-the-art performance across reliability, generalization, and locality metrics, scaling to thousands of sequential edits with minimal forgetting.

**Questions:**

Questions:

1. The paper focuses on factual knowledge editing. An interesting extension to the studied question is how would MEMOIR perform on other types of model updates, such as stylistic changes, or ethical alignment which are more subtle than facts and not necessarily true/false.

2. How does the fixed random permutation in TopHash affect the computational overhead during editing and inference, and is there a trade-off between the randomness introduced and the speed of the operation?

**Ethical Concerns:**

["NO or VERY MINOR ethics concerns only"]

**Limitations:**

Please refer to Strengths and Weaknesses.

**Quality:**

3

**Strengths And Weaknesses:**

Strengths:

1. The paper is overall well written, well motivated with good experimentation coverage.

2. The method's ability to dynamically activate or deactivate relevant knowledge based on prompt type (edited, rephrased, or irrelevant) effectively eliminates the need for large corpora of irrelevant samples during editing, which is often required by prior approaches to enhance locality. This is a strong differentiation with practical applications in real life.

3. The framework demonstrates strong scalability, extending the state-of-the-art edit horizon to 7000 edits while delivering superior performance in challenging sequential singular edit settings.

Weaknesses:

1. The performance of MEMOIR is sensitive to the number of active indices k used in TopHash. It appears crucial for balancing the model's ability to capture edits, prevent excessive overwriting, and maintain the quality of semantic relevance for routing. Although authors discuss it during the ablation, hyperparameter tuning aspect of this parameter k might be challenging in practice.

2. While the author claim that MEMOIR is scalable, the computational efficiency and memory footprint for extremely large LLMs (beyond 8B parameters) or significantly more edits than 7000 are not explicitly detailed. However, the method's design with a residual memory and sparse updates suggests better scaling than full fine-tuning.

3. The authors acknowledge that the paper does not present a formal theoretical results or proofs, focusing instead on empirical validation. While empirical results are strong, a theoretical foundation could further solidify the understanding of why MEMOIR performs well.

---

> ### Author Rebuttal · Authors · 2025-07-31
>
> **General comment**
>
> First, we would like to respectfully bring to the reviewer’s attention a correction in our reported results due to a technical oversight. Specifically, when computing TopHash indices during inference, we inadvertently included label information by using features of the concatenated [prompt, label] sequence.
>
> We have corrected this by recomputing TopHash indices using centered features averaged over prompt tokens only. This affected the semantic representation used for similarity computations and the subsequent routing step, impacting mainly the generalization metric for ZsRE QA task with LLaMA 3. In contrast, reliability and locality are only minimally affected. We see no or negligible degradation in the remaining combinations of benchmarks and models across all metrics.
>
> The revised results are provided below. Importantly, none of our previous conclusions are affected. In particular, in all settings, MEMOIR performance **remains state-of-the-art**, consistently outperforming all baselines across all metrics with a strong margin over the second-best methods.
>
> **Table 1: Q&A task results.**
> |Method|T=1||||T=10||||T=100||||T=1000||||
> |-|-|-|-|-|-|-|-|-|-|-|-|-|-|-|-|-|
> ||Rel.|Gen.|Loc.|Avg.|Rel.|Gen.|Loc.|Avg.|Rel.|Gen.|Loc.|Avg.|Rel.|Gen.|Loc.|Avg.|
> |LLaMA-3-8B|
> |MEMOIR(before)|1.00|0.98|1.00|0.99|0.99|0.96|1.00|0.98|0.97|0.94|1.00|0.97|0.95|0.91|1.00|0.95|
> |GRACE|1.00|0.46|1.00|0.82|1.00|0.42|1.00|0.81|1.00|0.39|1.00|0.80|1.00|0.37|1.00|0.79|
> |WISE|0.92|0.84|1.00|0.92|0.78|0.74|0.98|0.83|0.62|0.60|1.00|0.74|0.66|0.64|1.00|0.77|
> |AlphaEdit|0.98|0.89|1.00|0.96|0.93|0.85|0.98|0.92|0.91|0.79|0.94|0.88|0.84|0.77|0.56|0.72|
> |**MEMOIR**|1.00|1.00|1.00|**1.00**|0.97|0.89|1.00|**0.95**|0.96|0.89|1.00|**0.95**|0.94|0.85|1.00|**0.93**|
> |Mistral-7B|
> |MEMOIR(before)|1.00|0.93|1.00|0.98|0.98|0.91|1.00|0.96|0.96|0.89|1.00|0.95|0.93|0.87|1.00|0.93|
> |GRACE|1.00|0.36|1.00|0.79|1.00|0.15|1.00|0.72|1.00|0.15|1.00|0.72|1.00|0.02|1.00|0.67|
> |WISE|0.98|0.97|1.00|0.98|0.92|0.89|1.00|0.94|0.87|0.80|1.00|0.89|0.70|0.67|1.00|0.79|
> |AlphaEdit|0.83|0.77|1.00|0.87|0.87|0.75|0.99|0.87|0.86|0.74|0.95|0.85|0.85|0.72|0.68|0.75|
> |**MEMOIR**|1.00|0.99|1.00|**1.00**|0.97|0.94|1.00|**0.97**|0.95|0.91|1.00|**0.95**|0.94|0.89|1.00|**0.94**|
>
> **Table 2: Hallucination correction task results.**
> ||LLaMA-3-8B |||||||| Mistral-7B ||||||||
> |-|-|-|-|-|-|-|-|-|-|-|-|-|-|-|-|-|
> ||T=1||T=10||T=100||T=600||T=1||T=10||T=100||T=600||
> |Method|Rel.|Loc.|Rel.|Loc.|Rel.|Loc.|Rel.|Loc.|Rel.|Loc.|Rel.|Loc.|Rel.|Loc.|Rel.|Loc.|
> |MEMOIR(before)|1.00|1.00|1.01|1.00|1.09|1.00|1.37|1.00|1.00|1.00|1.02|1.00|1.09|1.00|1.22|1.00|
> |GRACE|1.05|1.00|7.10e1|1.00|7.12e1|1.00|7.73e1|1.00|1.39|1.00|5.97|1.00|9.53|1.00|9.57|1.00|
> |WISE|4.93e1|0.98|1.46|0.95|2.10|0.99|3.20|0.99|1.40|1.00|2.56|0.94|1.31|0.99|5.21|0.93|
> |AlphaEdit|1.58|1.00|3.12|0.98|5.97|0.93|8.49e3|0.05|1.75|1.00|1.76|1.00|2.87|0.98|1.70e2|0.88|
> |**MEMOIR**|1.00|1.00|1.01|1.00|1.07|1.00|1.25|1.00|1.00|1.00|1.02|1.00|1.09|1.00|1.22|1.00|
>
>
>
> **Rebuttal**
>
> We thank the reviewer for their valuable feedback. We appreciate their recognition that our framework is both scalable and achieves superior performance in sequential edits, and we are happy that they found the paper well-written, the motivation clear, and the experiments comprehensive. Below, we address their specific comments in detail.
>
> **W1: Choice of the number of active indices $k$**
> In the framework of MEMOIR, the choice for the number of active indices $k$ results in a trade-off, i.e., the higher $k$, the easier to capture individual edits; the lower $k$, the less forgetting due to reduced interference with previous edits. Thus optimal $k$ in our experiments is chosen based on the average reliability after a sequence of edits. However, we note that the choice of $k$ is globally robust to different settings. For example, we use the same $k=4096$ for different models (LLaMA-2, LLaMA-3, and Mistral), different tasks (ZsRE QA dataset, and SelfCheckGPT hallucination-correction task), and different numbers of edits (from a single edit to a total of 7000 edits). This demonstrates strong robustness of the choice of $k$ across testing scenarios. Furthermore, we also showed in Figure 6 that the performance of MEMOIR is not sensitive to the choice of $k$, with the reliability metric remaining high across a wide range of $k$ (from 1024 to 8192). Therefore, while $k$ is an important hyperparameter in the framework, MEMOIR's performance remains stable across a broad range of values, and we expect hyperparameter tuning to be straightforward in practical settings.
>
> **W2: Memory footprint and scalability to large LLMs and number of edits**
> We thank the reviewer for highlighting the computational efficiency of our method. In the revised version, we will include a dedicated section in the Appendix discussing the computational and memory footprint of MEMOIR.
>
> In terms of **memory footprint**, MEMOIR only requires a copy of a single linear layer in an LLM and one lightweight binary (rather than float32) mask per edit. Specifically, for a LLaMA-3-8B model and 7000 edits, this means a memory increase of 0.7% and 0.1% for parameters and masks, respectively, resulting in an overall increase of 0.8%. Similarly, for a LLaMA-3-70B, it will have an increase of 0.3% for parameters and less than 0.05% for masks. Importantly, we see that scale favors MEMOIR's memory footprint, as larger models result in a decreased (percentage-wise) memory overhead.
>
> In terms of **computational efficiency**, MEMOIR is more efficient than prior methods that perform parameter updates across multiple layers (e.g., MEMIT, AlphaEdit), as it only modifies a single MLP projection layer, while still achieving higher performance.
>
> To further illustrate the scalability of MEMOIR to a large number of edits, we conducted additional evaluations of MEMOIR with 11k and 15k edits on ZsRE QA dataset. As shown in the table, MEMOIR maintains stable performance with no collapse up to 15k edits. These experiments showcase the superior robustness of MEMOIR to longer edits; while baselines suffer in one or even all metrics, MEMOIR maintains strong performance even for 15k edits.
>
> |Method|Rel.|Gen.|Loc.|Avg.|Rel.|Gen.|Loc.|Avg.|
> |-|-|-|-|-|-|-|-|-|
> ||11k||||15k||||
> |GRACE|0.99|0.28|1.00|0.76|0.99|0.28|1.00|0.76|
> |WISE|0.44|0.42|1.00|0.62|0.43|0.41|1.00|0.61|
> |AlphaEdit|0.01|0.01|0.00|0.01|0.02|0.02|0.00|0.01|
> |**MEMOIR**|0.89|0.81|0.99|**0.90**|0.87|0.78|0.98|**0.88**|
>
> **W3: Theory**
> As also acknowledged by the reviewer, MEMOIR is methodologically well-motivated and supported by strong empirical performance, including ablation studies that clarify the method’s design and its performance gains. It is further inspired by a long line of work on catastrophic forgetting [1,2,3], which provides a solid foundation for sparsely allocating memory to mitigate forgetting in lifelong learning scenarios.
> Formal analysis of LLMs remains challenging with current tools or requires strong assumptions that limit practical relevance. Following standard practice in knowledge editing (e.g., GRACE, WISE) and broader LLM research, we prioritize strong empirical evidence guided by intuition.
>
>
> **Q1: Other types of model updates**
> We thank the reviewer for the helpful suggestion. MEMOIR addresses a specific problem, i.e., factual knowledge editing in LLMs, building on prior work and focusing evaluation on modifying stored factual knowledge. While our motivation and experiments follow the knowledge editing literature, MEMOIR is a general framework that, in theory, can apply to any setting where modifying intermediate representations is beneficial. Following ROME’s insight that factual knowledge is stored in specific layers, subsequent knowledge editing methods (including ours) focus on one or a few layers. Similarly, prior work has shown that editing intermediate or final representations can be effective for stylistic control and ethical alignment [4,5]. As MEMOIR operates on intermediate layers, it shares similarities with these methods and we expect it to generalize to such applications.
>
> **Q2: Computational overhead**
> The fixed random permutation in our experiments introduces negligible computational overhead during both editing and inference. For instance, the fixed permutations and TopK computations in the TopHash algorithm take just 1.5 seconds out of a total 11-minute runtime, accounting for only **0.23%** of the total execution time. Given this negligible cost, there is no meaningful trade-off between the added randomness and computational efficiency.
>
> We hope this effectively addresses the reviewer’s concerns and remain available for any further discussion or clarification.
>
> [1] Dropout as an implicit gating mechanism for continual learning, Mirzadeh et al., CVPR 2020 Workshop.
>
> [2] PackNet: Adding Multiple Tasks to a Single Network by Iterative Pruning, Mallya et al., CVPR 2018.
>
> [3] Supermasks in Superposition, Wortsman et al., NeurIPS 2020.
>
> [4] Style Vectors for Steering Generative Large Language Models, Konen et al., EACL2024
>
> [5] Spectral Editing of Activations for Large Language Model Alignment, Qiu et al., NeurIPS 2024

---

### Decision · Program_Chairs · 2025-09-17

**Decision:**

Accept (poster)

**Comment:**

This paper studies lifelong model editing for large language models, where targeted facts are updated in language models multiple times in a row. The authors propose a novel method and demonstrate that it outperforms a large set of baselines on benchmark tasks across when editing multiple language models. The work was well-received by the reviewers, who appreciated the problem as important and hard, the solution as innovative and interesting, and also found the paper to be well-written and clear. The main negatives were largely acknowledged and resolved during the discussions, though I urge the authors to incorporate some of the identified weaknesses into the paper (e.g., editing just one layer). During the rebuttal period, the authors also identified and resolved a bug that led to new results, though the main findings remain true. Overall, I recommend accepting the paper because the strengths outweigh the weaknesses and will hopefully drive further work in this important area.